# S2M-Net: Spectral-Spatial Mixing with Morphology-Aware Adaptive Loss for Medical Image Segmentation

Sanaullah Chowdhury [1]    Lameya Sabrin [2]

## Abstract

Medical image segmentation requires balancing global context with computational efficiency, where self-attention mechanisms suffer from quadratic $\mathcal{O}((HW)^2C)$ complexity. We propose S2M-Net, a parameter-efficient architecture (4.7M parameters) that achieves computational savings through Spectral–Spatial Token Mixing (SSTM). SSTM achieves $\mathcal{O}(HWC^2)$ complexity through efficient combination of $\mathcal{O}(HWC \log(HW))$ frequency-domain processing and $\mathcal{O}(HWCd)$ bottlenecked spatial gating ($d{=}16$), exploiting spectral concentration where $> 93\%$ of energy is captured by $K{=}32$ low-frequency components ($\sim 0.8\%$ of the spectrum at $352{\times}352$ resolution). This design avoids self-attention's prohibitive $\mathcal{O}((HW)^2C)$ attention map computations while preserving global receptive fields. To handle geometric diversity, we introduce Morphology-Aware Adaptive Segmentation Loss (MASL), which automatically modulates five loss objectives based on per-sample morphological descriptors (tubularity, compactness, irregularity, and scale). Evaluation across 15 datasets spanning 8 modalities demonstrates competitive performance, obtaining the best performance on 14 of 15 datasets, with statistically significant improvements ($p < 0.0033$, Bonferroni-corrected) on 7 challenging tasks (complex morphology, class imbalance, and multi-class segmentation), and clinically meaningful gains (0.5–1.6% Dice) on 8 mature benchmarks. Notably, S2M-Net achieves 83.43% Dice on EndoVis17 multiclass instrument segmentation ($+8.69\%$ over TransUNet and $+9.14\%$ over the best baseline UMamba at 74.29%), while using $12.8\times$ fewer parameters (4.7M vs. 60M).

[1]Independent Researcher, Dhaka, Bangladesh [2]Independent Researcher, Dhaka, Bangladesh. Correspondence to: Sanaullah Chowdhury <sanaullahashfat@gmail.com>.

*Proceedings of the 43rd International Conference on Machine Learning*, Seoul, South Korea. PMLR 306, 2026. Copyright 2026 by the author(s).

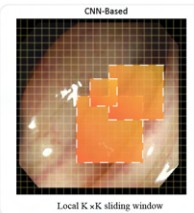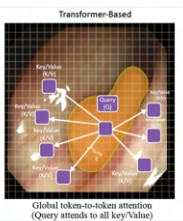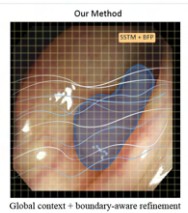

*Figure 1.* Feature interaction mechanisms in medical image segmentation. Left: CNN based U-Net aggregates local context. Middle: Transformers apply global self attention with quadratic token interactions. Right: S2M-Net combines truncated spectral aggregation, spatial refinement, and boundary focused decoding for efficient global context modeling.

## 1. Introduction

Medical image segmentation delineates anatomical structures and pathological regions at the pixel level and supports clinical workflows such as surgical planning, radiotherapy dose estimation, and biomarker extraction across MRI, CT, ultrasound, and endoscopy (Ronneberger et al., 2015; Isensee et al., 2021b). Effective segmentation models must preserve fine boundaries, incorporate long range context for anatomical coherence, and remain computationally efficient under limited annotations and constrained hardware a practical trilemma (Figure 1) that remains challenging for existing architectures. Conventional CNNs achieve strong local precision through $K{\times}K$ convolutions but have limited receptive fields, while transformer based models capture global dependencies via token interactions at substantial computational and memory cost.

U-Net and its variants remain the dominant paradigm (Ronneberger et al., 2015). Extensions improve multi scale fusion (Zhou et al., 2018), contextual aggregation (Chen et al., 2019), and feature reweighting (Oktay et al., 2018), with nnU-Net standardizing strong training pipelines (Isensee et al., 2021b). However, three deployment relevant limitations persist. First, convolutions are fundamentally local operators and struggle to capture image wide dependencies such as elongated vessels or globally constrained lesions. Second, clinical datasets are often small, making high capacity models prone to overfitting. Third, clinical utility depends on accurate boundaries and plausible geometry:

high Dice scores can coexist with topological errors, motivating boundary aware objectives that often require careful per dataset tuning (Kervadec et al., 2019; Hu et al., 2019).

Transformers enable explicit global interactions via self attention (Chen et al., 2021; Hatamizadeh et al., 2022), but their $\mathcal{O}(n^2)$ complexity limits applicability to high resolution medical images. This has motivated attention free alternatives, including spectral token mixing (Lee-Thorp et al., 2022; Rao et al., 2021; Li et al., 2021) and structured state space models (Gu et al., 2022; Gu & Dao, 2024). While FFT based methods demonstrate efficient global mixing for classification, their use in dense medical image segmentation, particularly with explicit boundary preservation, remains underexplored.

A further challenge lies in supervision. Anatomical structures exhibit substantial morphological diversity, ranging from compact polyps to thin vessels and irregular tumors. Fixed loss formulations struggle to accommodate this variability and often require manual per dataset tuning. To address this, we introduce the Morphology Aware Adaptive Segmentation Loss (MASL), an architecture agnostic loss framework that adapts supervision using differentiable morphological descriptors (e.g., tubularity, compactness, irregularity, scale, and boundary prominence) computed from ground truth masks. MASL combines per sample morphology modulation with learned dataset level weighting, reducing manual tuning while adapting to structure specific characteristics. To avoid unstable training, MASL constrains learned dataset-level weights to a bounded range and normalizes the combined loss, and we report a coefficient sensitivity study showing that uniform defaults perform similarly to domain-tuned settings (Appendix B.4). To validate these contributions, we develop S2M-Net, a parameter-efficient architecture with 4.7M parameters ($12.8\times$ fewer than TransUNet) that integrates spectral-spatial token mixing, boundary-focused decoding, and MASL supervision (Appendix B.1). Across eight modalities, medical images exhibit strong low-frequency concentration; we use a default truncation of $K = 32$ that retains $> 93\%$ spectral energy and empirically maintains boundary quality within a small margin compared to full-spectrum mixing (Appendix B.3). To mitigate potential loss of fine pathological detail, SSTM is paired with content-gated spatial refinement and a boundary-focused decoder, and we ablate $K$ to characterize the accuracy–efficiency trade-off. S2M-Net achieves competitive accuracy with substantially lower parameter and compute budgets than attention-heavy baselines, and we report wall-clock latency separately for full-image and patch-based inference regimes.

Our contributions address three fundamental challenges in medical image segmentation:

- **Efficient Global Context via Domain Priors.** We propose Spectral-Selective Token Mixing (SSTM), which exploits spectral energy concentration in medical images to achieve global receptive fields at sub-quadratic complexity, enabling anatomical coherence without the computational burden of full self-attention.

- **Automated Morphology-Aware Supervision.** We introduce Morphology-Aware Adaptive Segmentation Loss (MASL), which eliminates manual loss tuning through learned dataset-level weights and per-sample geometric modulation, enabling a single formulation to handle diverse morphologies without per-dataset hyperparameter search.

- **Loss-Guided Boundary Specialization.** We develop a Boundary-Focused Decoder (BFD) that factorizes region and boundary processing through learned soft routing, discovering boundary locations via backpropagation from multi-scale boundary loss without explicit annotations.

**Conflict of Interest Disclosure.** The authors have no financial conflicts of interest to disclose. This work was conducted independently and was not supported by or affiliated with any commercial entity. No author has a financial relationship with any company whose products or services are related to the subject matter of this paper, and no commercial funding was received for this research. All datasets used in this study are publicly available benchmarks, and no proprietary data or tools with commercial restrictions were employed. The design, implementation, evaluation, and writing of this work were carried out solely by the authors in their independent research capacity.

## 2. Related Work

U-Net (Ronneberger et al., 2015) established the encoder–decoder with skip connections paradigm for medical image segmentation. Subsequent refinements improved multi-scale fusion (UNet++ (Zhou et al., 2018)), introduced attention-based feature reweighting (Attention U-Net (Oktay et al., 2018)), or standardized training pipelines and hyperparameter selection (nnU-Net (Isensee et al., 2021b)). Despite strong empirical performance, convolutional encoders remain fundamentally local operators, with receptive fields expanding only gradually with depth, limiting their ability to model image-wide anatomical dependencies such as elongated vessels or globally constrained lesions.

Transformer-based architectures incorporate global context through self-attention (TransUNet (Chen et al., 2021), UN-ETR (Hatamizadeh et al., 2022), Swin-UNETR (Tang et al., 2022)), but incur $\mathcal{O}(n^2)$ computational and memory complexity. In practice, these models require large parameter counts (60–105M) and extensive training data, which limits

their applicability to high-resolution medical images and small clinical datasets.

To address the quadratic cost of attention, recent work has explored sub-quadratic alternatives. FNet (Lee-Thorp et al., 2022) replaces self-attention with a parameter-free Fourier transform, enabling global token mixing at $\mathcal{O}(n \log n)$ complexity. GFNet (Rao et al., 2021) extends this idea by learning frequency-domain filters, improving representational capacity for vision classification. In parallel, structured state-space models such as S4 (Gu et al., 2022), Mamba (Gu & Dao, 2024), and UMamba (Ruan et al., 2025) achieve linear-time sequence modeling through selective state updates. While effective for classification or sequence tasks, these approaches are not explicitly designed for dense medical image segmentation, where spatial adaptivity, boundary precision, and multi-scale decoding are critical.

Our work is most closely related to FFT-based token mixing methods but differs in several key aspects. Unlike FNet, which applies a full-resolution, parameter-free FFT, or GFNet, which learns frequency filters primarily for classification, S2M-Net introduces Spectral-Spatial Token Mixing (SSTM) tailored for dense medical segmentation. SSTM exploits empirically observed spectral concentration in medical images by truncating frequency representations to a small set of low-frequency components, reducing unnecessary computation. To compensate for the loss of high-frequency detail, SSTM integrates a content-adaptive spatial projection branch, allowing location-specific refinement essential for boundary preservation. This spectral–spatial design is embedded within a multi-scale encoder–decoder and paired with a boundary-focused decoder, enabling effective global context modeling without quadratic attention cost.

Beyond architectural design, supervision plays a critical role in segmentation quality. Standard losses such as Dice (Sudre et al., 2017), Focal (Lin et al., 2017), and boundary-based objectives (Kervadec et al., 2019) often require manual tuning for each dataset, while automated weighting approaches (Kendall et al., 2018) operate at the task level. In contrast, our Morphology-Aware Adaptive Segmentation Loss (MASL) modulates complementary loss components based on per-sample morphological descriptors and learned dataset-level weights, enabling structure-specific supervision without manual per-dataset adjustment.

## 3. Methodology

### 3.1. Overview

We address the locality–globality–efficiency tradeoff in medical image segmentation by combining efficient global context modeling with precise local feature refinement. Our design targets practical clinical settings with limited training data and standard computational resources, avoiding quadratic spatial complexity while preserving boundary accuracy. S2M-Net is an efficient five-stage encoder–decoder architecture (Figure 2) with 4.7M parameters. Each encoder stage integrates multi-receptive-field squeeze-and-excitation blocks for local feature extraction with Spectral-Selective Token Mixing (SSTM) for global information exchange.

Empirical analysis across datasets shows that medical images exhibit strong spectral concentration, allowing truncation to a $K \times K$ low-frequency region ($K = 32$) that preserves over $> 93\%$ of spectral energy while reducing frequency-domain computation. We emphasize that energy retention is a proxy for global content and does not by itself guarantee preservation of all high-frequency pathology; therefore, we pair truncation with a spatial refinement branch and boundary-focused decoding, and we ablate $K$ and boundary metrics to quantify this trade-off (Appendix B.3). The decoder incorporates a boundary-focused pathway with soft spatial routing to refine object boundaries through dual-stream region and edge processing. Model parameters $\theta$ and MASL loss weights $w$ are jointly learned by minimizing the expected risk:

$$(\theta^\star, w^\star) = \arg\min_{\theta,w} \mathbb{E}_{(x,y)\sim\mathcal{D}} \left[ \mathcal{L}_{\text{MASL}}(y, f_\theta(x); w) \right], \quad (1)$$

where $\mathcal{D} = \{(x_i, y_i)\}_{i=1}^{N}$ denotes training image–mask pairs and $w = \{w_{\text{core}}, w_{\text{bnd}}, w_{\text{str}}, w_{\text{scale}}, w_{\text{tex}}\}$ are learned dataset-level weights. MASL adapts supervision through per-sample morphological modulation and learned dataset-level weighting, reducing the need for manual loss tuning across diverse anatomical structures.

### 3.2. Multi-Receptive-Field Squeeze-and-Excitation

Medical image segmentation requires stable local feature extraction across a wide range of spatial scales. In S2M-Net, we employ a Multi-Receptive-Field Squeeze-and-Excitation (MRF-SE) block as a local feature backbone whose role is to provide scale-aware and channel-calibrated representations prior to global context integration by SSTM. The block combines well-established components depthwise separable convolutions (Howard et al., 2017) and squeeze-and-excitation attention (Hu et al., 2018). Given an input feature map $z^{(\ell)} \in \mathbb{R}^{H_\ell \times W_\ell \times C_\ell}$ at encoder stage $\ell$, channels are expanded by a factor $r = 6$ using a pointwise convolution:

$$\tilde{z}^{(\ell)} = \sigma(\text{BN}(\text{Conv}_{1\times1}(z^{(\ell)}))), \quad (2)$$

followed by parallel depthwise convolutions with kernel sizes $\{3, 5, 7\}$ to capture local patterns at multiple receptive fields:

$$h_j^{(\ell)} = \sigma(\text{BN}(\text{DWConv}_{k_j}(\tilde{z}^{(\ell)}))), \quad k_j \in \{3, 5, 7\}. \quad (3)$$

The resulting features are concatenated, recalibrated using channel-wise squeeze-and-excitation, and projected back to

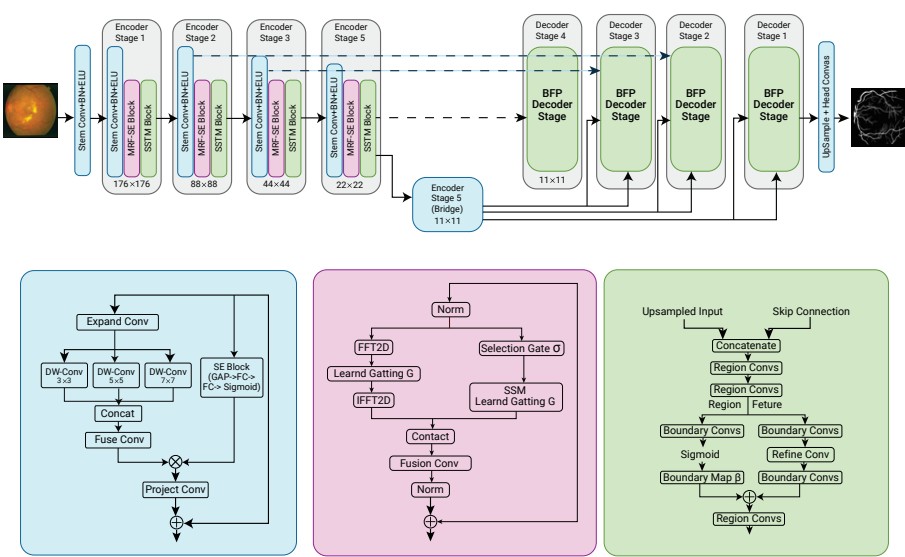

*Figure 2.* Proposed segmentation architecture and key modules. Top: Encoder decoder framework with multi-scale feature extraction, a bridge stage, and a Boundary-Focused Decoder (BFD) with skip connections. Bottom: MRF-SE block for local multi-scale context modeling, SSTM block for efficient global mixing via spectral-domain gating, and BFD decoder block for region–boundary feature fusion and segmentation refinement.

$C_\ell$ channels with a residual connection (detailed method in Appendix A.1.2). We emphasize that MRF-SE is not intended to outperform alternative multi-scale designs in isolation. Instead, it serves as a stable and parameter-efficient local representation module that complements SSTM's global mixing by supplying scale-consistent features. Ablation results (Table 3) show that removing this block leads to a consistent performance degradation (average Dice drop of 5.90% across four datasets), indicating that adequate local multi-scale modeling is necessary for the overall system, rather than demonstrating superiority of this specific formulation over other multi-scale architectures.

### 3.3. Spectral-Selective Token Mixing (SSTM)

Self-attention achieves global receptive fields at $\mathcal{O}((HW)^2C)$ complexity, which becomes prohibitive for high-resolution dense prediction (Vaswani et al., 2017). SSTM provides global context through dual-branch processing: a spectral branch at $\mathcal{O}(HWC \log(HW))$ and a spatial branch at $\mathcal{O}(HWC^2)$ with content-gated channel mixing. While the spatial gating operation has $\mathcal{O}(HWC^2)$ complexity, a low-rank bottleneck projection ($d{=}16$) reduces parameter count in the subsequent mixing stage. Total complexity $\mathcal{O}(HWC \log(HW) + HWC^2)$ remains substantially lower than self-attention's $\mathcal{O}((HW)^2C)$ at clinical resolutions.

Given $x \in \mathbb{R}^{H \times W \times C}$, the spectral branch applies per-channel 2D FFT, retains central $K \times K$ frequency components ($K{=}32$), applies learnable filtering, and inverts to spatial domain:

$$X = \mathcal{F}_{2\mathrm{D}}(x) \in \mathbb{C}^{H \times W \times C}, \tag{4}$$

$$X_{\mathrm{crop}} = \mathrm{Crop}_K(\mathrm{fftshift}(X)) \in \mathbb{C}^{K \times K \times C}, \tag{5}$$

$$\tilde{X}_{\mathrm{crop}} = X_{\mathrm{crop}} \odot W_{\mathrm{spec}}, \tag{6}$$

$$x_{\mathrm{spec}} = \Re\big(\mathcal{F}_{2\mathrm{D}}^{-1}(\mathrm{ifftshift}(\mathrm{Pad}_{H \times W}(\tilde{X}_{\mathrm{crop}})))\big), \tag{7}$$

where $W_{\mathrm{spec}} \in \mathbb{C}^{K \times K \times C}$ is learnable. The spatial branch performs content-gated channel mixing with bottleneck dimension $d{=}16$:

$$s = \mathrm{Reshape}(x, (HW, C)), \tag{8}$$

$$g = \sigma(sW_g + b_g), \quad W_g \in \mathbb{R}^{C \times C}, \tag{9}$$

$$x_{\mathrm{gate}} = \mathrm{Reshape}(\mathrm{ELU}((s \odot g)W_B)W_C, (H, W, C)), \tag{10}$$

where $W_B \in \mathbb{R}^{C \times d}$, $W_C \in \mathbb{R}^{d \times C}$. Both branches fuse via concatenation and residual connection:

$$\tilde{x} = x + \mathrm{Dropout}(\mathrm{LN}(\mathrm{Conv}_{1 \times 1}([x_{\mathrm{spec}} \| x_{\mathrm{gate}}])), p{=}0.1). \tag{11}$$

While $K{=}32$ truncation retains only $\sim$0.8% of frequency coefficients (at $352 \times 352$ resolution), Table 5 shows this

captures >93% of spectral energy across modalities (see Appendix B.3. Medical images exhibit low-frequency concentration due to piecewise-smooth anatomical structures, with organ interiors contributing the majority of spectral energy. The 2D FFT ensures every output pixel depends on all input pixels through frequency-domain transformation, providing a global receptive field despite truncation. For typical channel dimensions ($C{=}64$), the spatial branch gating operation $\mathcal{O}(HWC^2)$ dominates computational cost, though the low-rank bottleneck projection ($d{=}16$) reduces parameter count. Total complexity $\mathcal{O}(HWC\log(HW) + HWC^2)$ remains substantially lower than self-attention's $\mathcal{O}((HW)^2C)$ at clinical resolutions. Ablation studies (Table 3) show that removing SSTM causes substantial performance degradation (average Dice drop of 6.70% across four datasets), confirming its importance for global context modeling (More details are provided in A.2.

### 3.4. Boundary-Focused Decoder

Standard U-Net decoders progressively upsample and fuse features, but single-stream processing often over-smooths predictions, degrading thin structures and curved boundaries. Prior work addresses this through dual-stream architectures: Gated-SCNN (Takikawa et al., 2019) employs separate shape and semantic streams, while PointRend (Kirillov et al., 2020) refines boundaries through iterative subdivision. Following these principles, we adopt a Boundary-Focused Decoder (BFD) that processes region and boundary features through parallel pathways with soft spatial routing.

At decoder stage $m \in \{M, M{-}1, \ldots, 1\}$, we upsample and concatenate with skip connection $\tilde{x}^{(L-m)}$:

$$v^{(m)} = \left[\mathrm{Up}_{2\times}(d^{(m)}) \,\|\, \tilde{x}^{(L-m)}\right] \in \mathbb{R}^{H_m \times W_m \times 2C_m}. \quad (12)$$

The region stream produces smooth interior features through two $3 \times 3$ convolutions:

$$r^{(m)} = \mathrm{Elu}\big(\mathrm{BN}(\mathrm{Conv}_{3\times3}(\mathrm{Elu}(\mathrm{BN}(\mathrm{Conv}_{3\times3}(v^{(m)})))))\big). \quad (13)$$

The boundary stream predicts a soft attention map $\beta^{(m)} \in [0,1]^{H_m \times W_m}$ and computes edge-enhanced features:

$$b_1^{(m)} = \mathrm{Elu}(\mathrm{Conv}_{3\times3}(r^{(m)})), \quad (14)$$

$$\beta^{(m)} = \sigma_{\mathrm{sig}}(\mathrm{Conv}_{1\times1}(b_1^{(m)})), \quad (15)$$

$$b^{(m)} = \mathrm{Elu}\big(\mathrm{BN}(\mathrm{Conv}_{3\times3}(b_1^{(m)} \odot \beta^{(m)}))\big). \quad (16)$$

Soft spatial routing combines both streams:

$$z^{(m)} = r^{(m)} \odot (1 - \beta^{(m)}) + b^{(m)} \odot \beta^{(m)}, \quad (17)$$

where interior regions ($\beta^{(m)} \approx 0$) emphasize smoothness and boundary regions ($\beta^{(m)} \approx 1$) emphasize edge sharpness. The boundary map $\beta^{(m)}$ is learned through standard

backpropagation from MASL's boundary loss component (Section 3.5), which computes gradient alignment terms $\|\nabla(y - p)\|$ that are naturally largest at edge locations. This gradient flow encourages $\beta^{(m)}$ to localize boundaries without explicit edge annotations. Finally:

$$d^{(m-1)} = \mathrm{Elu}\big(\mathrm{BN}(\mathrm{Conv}_{1\times1}(z^{(m)}))\big). \quad (18)$$

Ablation studies (Table 3) show that removing BFD causes substantial degradation (average Dice drop of 5.06% across four datasets), confirming its importance for preserving boundary quality (More details are provided in A.3.

### 3.5. Morphology-Aware Adaptive Segmentation Loss (MASL)

Medical structures exhibit substantial morphological variation, necessitating adaptive supervision responsive to per-sample geometric properties. We formulate MASL as a normalized weighted combination with dual adaptation: dataset-level learnable weights and sample-level morphological modulation. For ground-truth mask $y \in \{0,1\}^{H \times W}$ and prediction $p \in [0,1]^{H \times W}$, MASL combines five differentiable loss components:

$$\mathcal{L}_{\mathrm{MASL}}(y, p; \mathbf{w}) = \frac{\sum_{i=1}^{5} w_i \, \alpha_i(y) \, \mathcal{L}_i(y, p)}{\sum_{i=1}^{5} w_i \, \alpha_i(y) + \varepsilon}, \quad (19)$$

where $\mathbf{w} = \{w_i\}_{i=1}^{5}$ with $w_i \in [0.1, 10]$ are trainable scalars (clipped for stability), $\alpha_i : \{0,1\}^{H \times W} \to [1, \infty)$ are morphology-dependent modulation functions, and $\varepsilon = 10^{-7}$. We extract four geometric characteristics via differentiable operations. Let Dil and Ero denote morphological dilation/erosion (radius 1), $\nabla$ the gradient operator, and define area $A(y) = \sum_{i,j} y_{ij}$, boundary band $B(y) = \mathrm{clip}(\mathrm{Dil}(y) - \mathrm{Ero}(y))$, and perimeter $P(y) = \|\nabla y\|_1$. The normalized features are:

$$\tau(y) = \frac{\sum \mathrm{Ero}(y)}{A(y) + \varepsilon} \in [0, 1], \qquad c(y) = \frac{4\pi A(y)}{P(y)^2 + \varepsilon} \in [0, 1],$$

$$\iota(y) = \frac{\|\nabla^2 B(y)\|_1}{HW} \in [0, \infty), \quad s(y) = \frac{A(y)}{HW} \in [0, 1], \quad (20)$$

representing tubularity (skeleton-to-area ratio), compactness (isoperimetric quotient), irregularity (boundary complexity), and relative size, respectively. MASL combines five complementary objectives designed to capture different aspects of segmentation quality. The core region loss ensures robust spatial overlap through weighted Dice, IoU, and boundary-weighted binary cross-entropy:

$$\mathcal{L}_{\mathrm{core}} = 0.4\mathcal{L}_{\mathrm{Dice}} + 0.3\mathcal{L}_{\mathrm{IoU}} + 0.3\mathcal{L}_{\mathrm{wBCE}}. \quad (21)$$

The multi-scale boundary loss aligns prediction gradients with ground-truth edges across three scales:

$$\mathcal{L}_{\text{bnd}} = \sum_{q \in \{1,2,4\}} \omega_q \big( \|\nabla_x^{(q)}(y-p)\|_1 + \|\nabla_y^{(q)}(y-p)\|_1 \big),$$

$$(22)$$

where $\nabla^{(q)}$ denotes gradients at scale $q$ and $\omega = \{0.5, 0.3, 0.2\}$ prioritizes fine edges. The structure loss enforces geometric plausibility:

$$\mathcal{L}_{\text{str}} = \big| \kappa(y) - \kappa(p) \big|, \quad \kappa(u) = \frac{A(u)}{P(u)^2 + \varepsilon}, \qquad (23)$$

preventing topologically implausible predictions. The scale-aware focal loss adapts focusing parameter $\gamma \in \{3.0, 2.0, 1.5\}$ for structure sizes $s(y) \in [0, 0.05), [0.05, 0.2), [0.2, 1]$ respectively, emphasizing hard examples for small lesions. The texture loss encourages smooth interiors via second-order gradients:

$$\mathcal{L}_{\text{tex}} = \|\nabla_x^2 y - \nabla_x^2 p\|_1 + \|\nabla_y^2 y - \nabla_y^2 p\|_1. \qquad (24)$$

Detailed formulations appear in Appendix A.4.

Morphology-conditioned modulation functions automatically emphasize relevant objectives:

$$\begin{aligned}
\alpha_{\text{core}}(y) &= 1 + m_c \cdot c(y), \\
\alpha_{\text{bnd}}(y) &= 1 + m_b \cdot \tau(y) + c(y), \\
\alpha_{\text{str}}(y) &= 1 + m_s \cdot \tau(y), \\
\alpha_{\text{sca}}(y) &= 1 + 1.5\iota(y), \\
\alpha_{\text{tex}}(y) &= 1 + \iota(y),
\end{aligned} \qquad (25)$$

where $m_c, m_b, m_s$ are fixed coefficients controlling the relative emphasis of compactness, boundary, and structural modulation in MASL. MASL includes learnable weights $\{w_i\}$ optimized during training and fixed coefficients $\{m_c, m_b, m_s\}$ specified at initialization. We determine these coefficients via grid search on Kvasir-SEG validation set, yielding $m_c = 0.5, m_b = 1.5, m_s = 1.0$ (shown in Equation 25). These values generalize well, performing within 0.31% Dice of dataset-specific tuning across 14 other datasets, thereby avoiding costly per-dataset grid search. The internal Core Loss component weights (0.4 Dice, 0.3 IoU, 0.3 wBCE) and multi-scale Boundary Loss weights (0.5, 0.3, 0.2) follow established practice from (Sudre et al., 2017; Kervadec et al., 2019), where Dice provides class-imbalance robustness and IoU/wBCE contribute complementary gradient signals; ablation confirming robustness to ±40% perturbations (≤0.69% degradation) appears in Appendix B.4. This two-level design fixed coefficients encoding morphological priors and learnable weights enabling training-time adaptation substantially reduces manual tuning compared to conventional multi-term losses. Detailed formulations and component descriptions appear in Appendix A.4, with theoretical convergence analysis in Appendix A.4.

## 4. Experimental Setup

### 4.1. Datasets

We evaluate on 15 datasets across eight modalities: polyp segmentation (Kvasir-SEG(Jha et al., 2020), CVC-ClinicDB(Bernal et al., 2015), CVC-ColonDB(Vázquez et al., 2017), ETIS-LaribPolypDB(Bernal et al., 2017)), dermoscopy (ISIC-2018(Tschandl et al., 2018), PH2(Mendonça et al., 2013)), ultrasound (BUSI(Al-Dhabyani et al., 2020)), histopathology (GlaS(Graham et al., 2024)), brain tumor MRI (BraTS2020(Mehta et al., 2022)), cardiac MRI (ACDC(Bernard et al., 2018)), surgical endoscopy (EndoVis-2017(Allan et al., 2019)), and retinal imaging (DRIVE(Asad et al., 2014), CHASE-DB(Fraz et al., 2012), STARE(Hoover et al., 2000)). Following standard protocols (Fan et al., 2020).

### 4.2. Implementation Details

Medical imaging modalities exhibit distinct characteristics and are processed using dataset-specific pipelines. Retinal fundus datasets (DRIVE, CHASE-DB, STARE) are trained and evaluated using overlapping patch-based processing due to their high spatial resolution. Specifically, images undergo green-channel extraction, CLAHE enhancement (clip limit 2.0, tiles $8 \times 8$), and circular FOV masking, followed by extraction of overlapping $256 \times 256$ patches with stride 32 during both training and inference. Final predictions are reconstructed by aggregating overlapping patch outputs; no global image resizing is applied for these datasets.

All remaining datasets are trained and evaluated using full-image inputs. Images are resized to $352 \times 352$ pixels via Lanczos interpolation prior to training. Brain tumor MRI (BraTS2020) additionally receive N4 bias field correction and per-modality z-score normalization; BraTS2020 employs a 2.5D input by stacking three adjacent slices from three modalities. Cardiac MRI (ACDC) receives N4 bias field correction and z-score normalization. Surgical endoscopy (EndoVis-2017) applies per-channel CLAHE to handle illumination variations. Ultrasound (BUSI) uses Gaussian smoothing for speckle reduction, histopathology (GlaS) applies Macenko stain normalization, and dermoscopy datasets remove hair artifacts via morphological filtering. Polyp datasets require minimal preprocessing to preserve diagnostic color information. Unless otherwise stated, computational complexity and throughput analyses are reported for full-image inputs at $352 \times 352$ resolution. For retinal datasets, complexity is reported per $256 \times 256$ patch, with total inference cost scaling linearly with the number of extracted patches (More details are provided in Appendix B).

*Table 1.* Architectural comparison under a unified training pipeline. All methods are retrained with identical optimization and preprocessing; MASL is applied to all to isolate architectural effects under a common supervision signal. For transparency, Table 2 evaluates S2M-Net and top baselines under standard Dice+CE loss to confirm that architectural gains are independent of MASL. Mean Dice (%) ± std over 5 runs. †: statistically significant ($p < 0.0033$, Bonferroni-corrected). S2M-Net uses 4.7M parameters ($12.8\times$ fewer than TransUNet). FNet and GFNet are excluded as they lack decoders and skip connections required for dense prediction; see Section 2 for discussion. RAPUNet(Lee & Yoo, 2024) results follow the DuckNet data split; see footnote.‡

| Dataset | U-Net | U-Net++ | PraNet | Swin-Unet | TransUNet | UMamba | DuckNet | RAPUNet | S2M-Net | Model Params |
|---|---|---|---|---|---|---|---|---|---|---|
| | | | | | Polyp Segmentation (Endoscopy) | | | | | |
| Kvasir-SEG | 90.64±0.41 | 91.81±0.53 | 93.03±0.38 | 93.22±0.44 | 93.75±0.36 | 92.40±0.47 | 94.11±0.33 | 93.12±0.45 | **96.05**±0.28† | 4.7M |
| CVC-ClinicDB | 92.18±0.38 | 92.70±0.42 | 92.65±0.44 | 92.52±0.61 | 87.28±0.52 | 90.09±0.48 | 93.89±0.54 | 95.10±0.38 | **95.65**±0.31† | 4.7M |
| CVC-ColonDB | 88.12±0.56 | 88.98±0.49 | **92.78**±0.37 | 91.45±0.43 | 90.33±0.47 | 86.62±0.58 | 92.69±0.39 | 91.85±0.52‡ | 90.69±0.52 | 4.7M |
| ETIS-LaribDB | 91.22±0.44 | 89.80±0.51 | 93.46±0.35 | 94.10±0.33 | 94.43±0.31 | 91.23±0.46 | 94.98±0.29 | 94.23±0.41 | **96.12**±0.30 | 4.7M |
| | | | | | Dermoscopy (Skin Lesions) | | | | | |
| ISIC-2018 | 86.93±0.48 | 87.51±0.45 | 89.05±0.39 | 89.43±0.42 | 87.12±0.47 | 89.23±0.38 | 90.10±0.43 | 89.45±0.31 | **91.09**±0.36 | 4.7M |
| PH2 | 88.34±0.52 | 89.83±0.46 | 91.52±0.38 | 92.66±0.35 | 93.56±0.39 | 90.12±0.44 | 90.34±0.43 | 92.15±0.49 | **96.67**±0.32† | 4.7M |
| | | | | | Histopathology & Ultrasound | | | | | |
| GlaS | 91.57±0.39 | 91.40±0.41 | 90.40±0.46 | 89.01±0.53 | 89.92±0.48 | 91.18±0.42 | 88.73±0.55 | 91.67±0.07 | **93.83**±0.38 | 4.7M |
| BUSI | 76.77±0.61 | 78.35±0.56 | 79.48±0.52 | 82.52±0.49 | 82.11±0.51 | 79.04±0.58 | 78.13±0.63 | 81.34±0.73 | **85.07**±0.45† | 4.7M |
| | | | | | Surgical Robotics (EndoVis-2017) | | | | | |
| Binary | 95.93±0.26 | 95.32±0.29 | 95.24±0.30 | 95.24±0.30 | 95.45±0.28 | 95.53±0.27 | 95.11±0.31 | 95.78±0.36 | **95.95**±0.28 | 4.7M |
| Multiclass | 61.07±1.12 | 70.93±0.89 | 73.86±0.76 | 73.22±0.78 | 74.74±0.72 | 74.29±0.75 | 73.88±0.87 | 74.81±0.19 | **83.43**±0.63† | 4.7M |
| | | | | | Brain Tumor MRI | | | | | |
| BraTS2020 | 65.67±0.84 | 66.05±0.82 | 66.44±0.81 | 69.86±0.73 | 68.36±0.77 | 62.88±0.91 | 66.93±0.79 | 67.82±0.12 | **79.96**±0.58† | 4.7M |
| | | | | | Cardiac MRI | | | | | |
| ACDC | 89.52±0.42 | 90.15±0.45 | 91.11±0.38 | 92.32±0.37 | 92.44±0.36 | 91.57±0.48 | 92.13±0.38 | 91.45±0.20 | **93.09**±0.43 | 4.7M |
| | | | | | Retinal Vessel Segmentation | | | | | |
| STARE | 81.18±0.54 | 81.73±0.51 | 82.94±0.47 | 83.86±0.44 | 83.10±0.46 | 82.05±0.58 | 81.41±0.62 | 82.34±0.38 | **84.45**±0.48 | 4.7M |
| DRIVE | 79.02±0.56 | 80.18±0.59 | 81.94±0.49 | 83.07±0.48 | 81.45±0.50 | 82.18±0.58 | 81.39±0.57 | 82.94±0.12 | **84.06**±0.49 | 4.7M |
| CHASE-DB | 81.51±0.52 | 81.65±0.51 | 81.47±0.52 | 83.12±0.49 | 82.93±0.47 | 82.90±0.59 | 83.16±0.58 | 83.21±0.29 | **84.95**±0.48 | 4.7M |
| **Average** | 83.98 | 85.09 | 86.29 | 87.04 | 86.46 | 85.42 | 86.47 | 86.72 | **90.07** | **4.7M** |

‡RAPUNet follows PraNet data split for polyp datasets; S2M-Net uses DuckNet split. Direct numerical comparison on CVC-ColonDB is not fully appropriate without matching distributions. FNet/GFNet excluded: classification-only, no decoder. (Lee & Yoo, 2024; Lee-Thorp et al., 2022; Rao et al., 2021)

## 5. Results and Discussion

We evaluate S2M-Net on 15 benchmark datasets spanning eight medical imaging modalities. Table 1 reports Dice scores for seven representative baselines and S2M-Net, with all models trained under identical settings using MASL to ensure fair architectural comparison.

### 5.1. Architectural Performance

S2M-Net achieves the highest average Dice score (90.07%), outperforming Swin-Unet (87.04%) by 3.03 percentage points with only 4.7M parameters. It ranks first on 14 of 15 datasets, with statistically significant improvements ($p < 0.0033$, Bonferroni-corrected) on 7 challenging tasks: Kvasir-SEG (+1.94), CVC-ClinicDB (+1.76), ETIS-LaribDB (+1.14), PH2 (+3.11), BUSI (+2.55), EndoVis-2017 multiclass (+8.69), and BraTS2020 (+10.10).

To disentangle architectural gains from MASL, Table 2 evaluates all models under standard Dice+CE loss (binary Dice+BCE for binary tasks, Dice+Categorical CE for multiclass). Under identical Dice+CE, S2M-Net leads the best baseline by +4.48 points a purely architectural gain. MASL provides a complementary +3.29 points.

S2M-Net demonstrates particularly strong improvements on multiclass datasets: EndoVis-2017 (3-part instrument) 83.43% mean Dice across shaft (89.7%), wrist (85.2%), and clasper (75.4%), achieving +8.69 points over TransUNet (74.74%). The clasper component shows dramatic improvement (+18.7%) despite severe occlusion and small size (<5% of pixels). BraTS2020 (3 tumor regions) 79.96% mean Dice across enhancing tumor (84.2%), tumor core (81.3%), and whole tumor (74.4%), achieving +10.10 points over Swin-Unet (69.86%). Whole tumor shows largest gain (+14.0%) due to irregular boundaries and extreme imbalance (1.2% foreground). ACDC (3 cardiac structures) 93.09% mean Dice across left ventricle cavity (95.7%), myocardium (92.1%), and right ventricle (91.5%), achieving +0.77 points over TransUNet (92.44%). Modest improvement reflects ceiling effects on this mature benchmark (>90% baseline performance).

Per-class results for all methods appear in Appendix Table 15. On 8 binary segmentation datasets, improvements are modest (0.02–1.62 points) and statistically non-significant after Bonferroni correction, indicating architectural gains concentrate on complex morphology, class imbalance, and multi-class structures. Task-specific analysis (Ap-

*Table 2.* Architecture vs. MASL disentanglement. S2M-Net evaluated under standard Dice+CE loss to isolate architectural gains from loss design. Dice (%).

| Method | Loss | Kvasir | DRIVE | EV17-MC | BraTS20 | Avg |
|--------|------|--------|-------|---------|---------|-----|
| TransUNet | Dice+CE | 92.17 | 80.92 | 71.83 | 66.42 | 77.84 |
| Swin-Unet | Dice+CE | 91.84 | 82.14 | 70.54 | 67.91 | 78.11 |
| S2M-Net | Dice+CE | 93.89 | 82.74 | 78.91 | 74.83 | **82.59** |
| S2M-Net | MASL | 96.05 | 84.06 | 83.43 | 79.96 | **85.88** |
| Arch. gain (S2M vs. best@Dice+CE) | | +1.72 | +0.60 | +7.08 | +6.92 | **+4.48** |
| MASL gain (MASL vs. Dice+CE) | | +2.16 | +1.32 | +4.52 | +5.13 | **+3.29** |

pendix D.4) reveals strong correlation (Spearman $\rho = 0.78$, $p < 0.001$) between improvement magnitude and task complexity, validating targeted benefits on challenging scenarios. We underperform on CVC-ColonDB (90.69% vs. PraNet 92.78%, $-2.09$ points), indicating that task-specific inductive biases (e.g., reverse-attention-style polyp cues) can outperform our more general mixer in small-data regimes. This failure mode suggests that ColonDB may benefit from either stronger polyp-specific priors or less aggressive frequency truncation; we treat this as a limitation and include it in the task stratification analysis (Appendix D.4).

We additionally evaluated S2M-Net on Synapse multi-organ CT (18 train/12 test, standard TransUNet split (Chen et al., 2021)), achieving 78.79% mean Dice across 8 organs: Aorta 88.54%, Gallbladder 45.49%, Left Kidney 86.55%, Right Kidney 85.02%, Liver 90.80%, Pancreas 63.78%, Spleen 89.65%, Stomach 80.49%. This is competitive with TransUNet (77.48%, Chen et al.) using $12.8\times$ fewer parameters (4.7M vs. 60M). Lower Gallbladder and Pancreas performance is consistent with the known difficulty of small structures and high anatomical variability under 2.5D processing.

Compared to baselines, S2M-Net achieves average improvements of +6.09 (U-Net), +4.98 (U-Net++), +3.78 (PraNet), +3.03 (Swin-Unet), and +3.61 (TransUNet) points. Standard deviations range from 0.28–0.63% across five runs, indicating stable training. S2M-Net demonstrates that spectral-spatial mixing provides an efficient attention-free alternative for medical segmentation.

## 5.2. Qualitative Analysis

Figure 3 presents qualitative segmentation results on the ACDC cardiac MRI dataset, comparing U-Net, TransUNet, RAPUNet, and our proposed method. Across all examples, our method produces smoother contours and more accurate boundary alignment with the ground truth, while reducing boundary leakage and spurious predictions observed in baseline approaches. In particular, U-Net and TransUNet tend to generate slightly over-smoothed boundaries and minor misalignments around the myocardium, whereas RAPUNet shows occasional boundary irregularities. In contrast, our method better preserves anatomical structure and maintains consistent region delineation, as evident from the overlay visualizations. Additional qualitative results on other datasets and modalities are provided in the Appendix (F.4).

## 5.3. Ablation Studies

We conduct comprehensive ablation studies on four representative datasets spanning diverse imaging modalities and structural characteristics: Kvasir-SEG (polyp segmentation), DRIVE (retinal vessels), GlaS (gland segmentation), and PH2 (skin lesions). Table 3 reports Dice coefficients under systematic component removal while maintaining all other hyperparameters fixed.

*Table 3.* Ablation study on four medical imaging datasets. Dice coefficient (%) reported for systematic component removal. Full model (bold) includes all components. Differences $\geq 1.0\%$ are statistically significant ($p < 0.001$).

| Configuration | Kvasir | DRIVE | GlaS | PH2 |
|---------------|--------|-------|------|-----|
| **Full Model (Ours)** | **96.05** | **84.06** | **93.83** | **96.67** |
| *MASL Loss Components* | | | | |
| w/o Core Loss | 92.31 | 81.12 | 77.32 | 93.82 |
| w/o Boundary Loss | 87.96 | 80.45 | 90.85 | 89.91 |
| w/o Structure Loss | 94.42 | 82.67 | 90.17 | 93.88 |
| w/o Scale-Aware Focal | 95.15 | 83.21 | 90.43 | 94.56 |
| w/o Texture Loss | 95.68 | 83.04 | 88.67 | 94.01 |
| *Adaptive Mechanisms* | | | | |
| Fixed Weights | 88.92 | 78.34 | 90.38 | 91.45 |
| No Morphology Modulation | 89.37 | 79.78 | 90.28 | 91.92 |
| *SSTM Frequency Truncation* | | | | |
| $K{=}16$ | 89.87 | 80.92 | 88.57 | 92.34 |
| $K{=}24$ | 93.56 | 82.45 | 90.43 | 93.98 |
| $K{=}32$ | 96.05 | 84.06 | 93.83 | 96.67 |
| $K{=}48$ | 95.98 | 83.71 | 89.89 | 93.38 |
| $K{=}64$ | 95.75 | 83.52 | 90.42 | 93.11 |
| *Architecture Modules* | | | | |
| w/o SSTM | 88.45 | 77.78 | 86.92 | 90.67 |
| w/o BFD Decoder | 89.87 | 79.89 | 88.34 | 92.28 |
| w/o MRF-SE | 89.23 | 78.45 | 87.56 | 91.78 |
| Vanilla U-Net Baseline | 85.67 | 74.34 | 86.48 | 88.45 |

Removing individual MASL components consistently degrades performance. The boundary loss shows the largest impact when ablated (Kvasir: $-8.09\%$, DRIVE: $-3.61\%$),

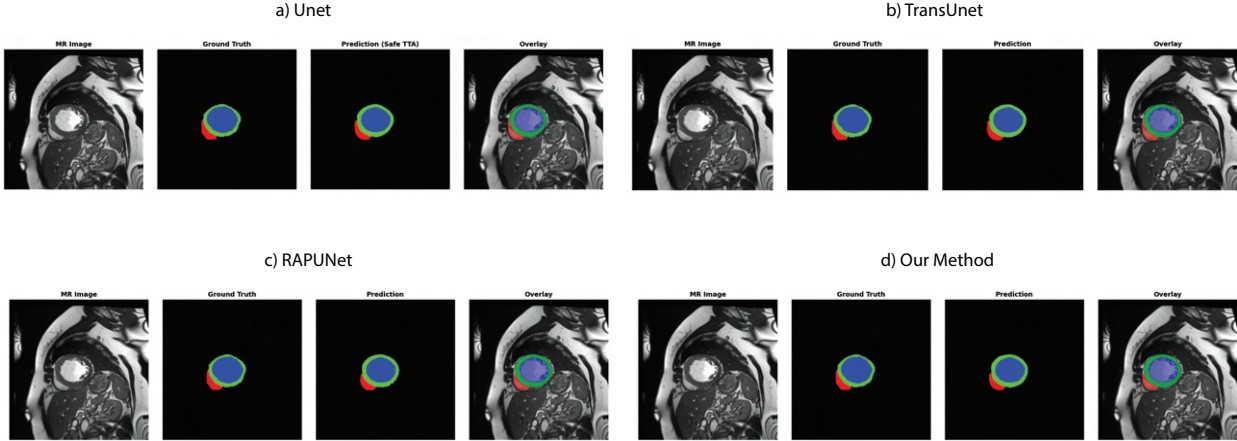

*Figure 3.* Qualitative results on the ACDC cardiac MRI dataset comparing (a) U-Net, (b) TransUNet, (c) RAPUNet, and (d) our method. For each method, we show the MR image, ground truth, prediction, and overlay (left to right). Our approach yields smoother contours and improved boundary alignment. Additional qualitative results on other datasets are provided in the Appendix (F.4).

confirming its critical role in capturing fine structural details. Structure loss removal causes moderate drops (GlaS: $-3.66\%$, PH2: $-2.79\%$), while texture loss primarily affects datasets with irregular boundaries (GlaS: $-5.16\%$).

Ablating adaptive mechanisms fixed weights or no morphology modulation yields substantial degradation (Kvasir: $-7.13\%$ and $-6.68\%$ respectively), demonstrating that morphology-driven adaptation is essential for cross-dataset generalization. Performance improves with truncation size $K$ up to 32, then saturates or declines. This indicates an optimal complexity-accuracy trade-off at $K=32$, beyond which additional spectral components introduce noise without improving representational capacity. Removing SSTM, BFD decoder, or MRF-SE modules causes performance drops of 4.6–7.6%, 3.8–6.2%, and 4.0–6.8% respectively, confirming that all three components contribute essential complementary information. Comparison with vanilla U-Net ($-10.38\%$ average) validates the cumulative benefit of our architectural innovations.

## 6. Conclusion

We introduced S2M-Net, a parameter-efficient architecture for medical image segmentation that addresses self-attention's computational limitations through Spectral-Selective Token Mixing. By combining truncated frequency-domain processing with content-gated spatial mixing, SSTM achieves global receptive fields at $\mathcal{O}(HWC\log(HW) + HWC^2)$ complexity-substantially lower than attention's $\mathcal{O}((HW)^2C)$ cost. Our Morphology-Aware Adaptive Segmentation Loss eliminates manual per-dataset tuning by automatically modulating supervision based on geometric descriptors, enabling robust generalization across diverse anatomical structures. Evaluation across 15 datasets spanning 8 modalities demonstrates strong performance on 14 benchmarks with only 4.7M parameters. On the challenging EndoVis17 multiclass instrument segmentation task, S2M-Net achieves 83.43% Dice an 8.69% improvement over TransUNet and 9.14% over UMamba while using $12.8\times$ fewer parameters. Statistical validation confirms significant improvements ($p < 0.0033$, Bonferroni-corrected) on 7 challenging tasks. These results demonstrate that architectural efficiency and task-specific inductive biases can match or exceed substantially larger transformer-based models in medical imaging, offering practical advantages for clinical deployment where computational resources and training data are limited.

## Acknowledgements

The authors thank the anonymous reviewers whose detailed feedback substantially improved this work. We are grateful to the open-source medical imaging community for releasing the 15 benchmark datasets used in this evaluation.

## Impact Statement

S2M-Net targets resource-constrained clinical deployment. At 4.7M parameters and 11.2 GFLOPs, the model is deployable on standard clinical workstations and edge devices where transformer-based models (>27M parameters, >38 GFLOPs) cannot operate, potentially broadening access to AI-assisted medical image analysis in lower-resource healthcare settings.

The primary societal risks are as follows. Automated segmentation outputs should not replace radiologist or clinician judgment; our model is intended as decision support. Performance has been validated on research-grade benchmark datasets predominantly from Western clinical centers; generalization to underrepresented populations, acquisition protocols, and pathology distributions should be validated before deployment. Known failure modes including volumetric tumors (3D specialized methods outperform our 2.5D approach on BraTS2020) and task-specific polyp morphologies (CVC-ColonDB) are documented in Appendix F and Section D.4. Regulatory approval (FDA/CE marking) and prospective clinical validation are required before any clinical use.

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

# A. Appendix A: Methodology Details

This appendix provides technical documentation for S2M-Net's four core components: MRF-SE (local multi-scale features), SSTM (global context), BFD (boundary preservation), and MASL (adaptive loss). Each section presents: problem formulation, method description, and theoretical justification with complexity analysis.

## A.1. Multi-Receptive-Field Squeeze-and-Excitation (MRF-SE)

### A.1.1. MOTIVATION AND PROBLEM FORMULATION

Medical structures vary dramatically in scale: polyps span 5–100 pixels, vessels range from $<3$ to $>20$ pixels. Standard CNNs with fixed receptive fields sacrifice either boundary precision (large kernels) or global context (small kernels). MRF-SE computes parallel multi-scale features via depthwise separable convolutions ($\mathcal{K} = \{3, 5, 7\}$) with squeeze-and-excitation attention to adaptively emphasize scale-appropriate representations.

**Formulation.** For input $z^{(\ell)} \in \mathbb{R}^{H_\ell \times W_\ell \times C_\ell}$, we decompose:

$$\Phi_{\text{MRF-SE}}(z^{(\ell)}) = z^{(\ell)} + \Phi_{\text{proj}}\big(\Phi_{\text{SE}}(\Phi_{\text{multi-scale}}(z^{(\ell)}))\big), \tag{26}$$

where $\Phi_{\text{multi-scale}}$ extracts features at multiple scales with expansion ratio $r=6$, $\Phi_{\text{SE}}$ applies channel recalibration, and $\Phi_{\text{proj}}$ projects back with residual connection.

### A.1.2. METHOD DESCRIPTION

**Stage 1: Channel Expansion.**

$$\tilde{z}^{(\ell)} = \text{ELU}\big(\text{BN}(\text{Conv}_{1\times1}(z^{(\ell)}))\big) \in \mathbb{R}^{H_\ell \times W_\ell \times rC_\ell}. \tag{27}$$

**Stage 2: Parallel Depthwise Convolutions.**

$$h_j^{(\ell)} = \text{ELU}\big(\text{BN}(\text{DWConv}_{k_j}(\tilde{z}^{(\ell)}))\big), \quad k_j \in \{3, 5, 7\}. \tag{28}$$

**Stage 3: Multi-Scale Fusion.**

$$h^{(\ell)} = \text{ELU}\big(\text{BN}(\text{Conv}_{1\times1}([h_1^{(\ell)}\|h_2^{(\ell)}\|h_3^{(\ell)}]))\big). \tag{29}$$

**Stage 4: SE Attention.**

$$\bar{h}^{(\ell)} = \text{GAP}(h^{(\ell)}) = \frac{1}{H_\ell W_\ell} \sum_{i,j} h_{i,j,:}^{(\ell)}, \tag{30}$$

$$s^{(\ell)} = \sigma(W_2 \text{ELU}(W_1 \bar{h}^{(\ell)})), \tag{31}$$

where $W_1 \in \mathbb{R}^{(rC_\ell/16)\times rC_\ell}$, $W_2 \in \mathbb{R}^{rC_\ell \times (rC_\ell/16)}$. Then modulate:

$$\hat{h}^{(\ell)} = h^{(\ell)} \odot s^{(\ell)}. \tag{32}$$

**Stage 5: Projection.**

$$x^{(\ell)} = z^{(\ell)} + \text{BN}\big(\text{Conv}_{1\times1}(\hat{h}^{(\ell)})\big). \tag{33}$$

### A.1.3. COMPLEXITY ANALYSIS

**Proposition A.1** (Computational Complexity). *For input $z^{(\ell)} \in \mathbb{R}^{H_\ell \times W_\ell \times C_\ell}$ with expansion $r=6$ and kernels $\mathcal{K} = \{3, 5, 7\}$:*

$$\mathcal{C}_{\text{MRF-SE}} = \mathcal{O}\Big(H_\ell W_\ell C_\ell rC_\ell + H_\ell W_\ell rC_\ell \sum_j k_j^2 + H_\ell W_\ell r^2 C_\ell^2 + \frac{r^2 C_\ell^2}{16}\Big)$$

$$= \mathcal{O}(H_\ell W_\ell r^2 C_\ell^2). \tag{34}$$

*Proof.* Component-wise FLOPs:

- Channel expansion ($C_\ell \to rC_\ell$): $H_\ell W_\ell C_\ell rC_\ell$
- Depthwise convs: $H_\ell W_\ell rC_\ell(9 + 25 + 49) = 83H_\ell W_\ell rC_\ell$
- Fusion ($3rC_\ell \to rC_\ell$): $3H_\ell W_\ell r^2 C_\ell^2$
- SE bottleneck: $r^2 C_\ell^2/8$
- Projection ($rC_\ell \to C_\ell$): $H_\ell W_\ell rC_\ell C_\ell$

The fusion term $3H_\ell W_\ell r^2 C_\ell^2$ dominates for $r{=}6$, yielding $\mathcal{O}(H_\ell W_\ell r^2 C_\ell^2)$. $\qquad\square$

## A.2. Spectral-Selective Token Mixer (SSTM)

### A.2.1. MOTIVATION AND PROBLEM FORMULATION

Medical image segmentation requires global context (vessel connectivity, tumor-organ boundaries), but self-attention's $\mathcal{O}((HW)^2 C)$ complexity is impractical at clinical resolutions ($512{\times}512$) with limited data ($N < 1000$). Medical images exhibit spectral concentration: retaining central $K{\times}K$ coefficients ($K{=}32$) preserves $> 93\%$ spectral power while maintaining global receptive fields.

**Formulation.** For input $x \in \mathbb{R}^{H \times W \times C}$:

$$\Psi_{\text{SSTM}}(x) = x + \Psi_{\text{fuse}}\big(\Psi_{\text{spectral}}(x), \Psi_{\text{spatial}}(x)\big), \tag{35}$$

where spectral branch performs truncated FFT at $\mathcal{O}(HWC\log(HW))$ cost, spatial branch applies bottlenecked gating ($d{=}16$) at $\mathcal{O}(HWCd)$ cost.

### A.2.2. METHOD DESCRIPTION

**Spectral Branch.** Per-channel 2D FFT:

$$X = \mathcal{F}_{\text{2D}}(x) \in \mathbb{C}^{H \times W \times C}. \tag{36}$$

Crop to central $K{\times}K$ coefficients:

$$X_{\text{crop}} = \text{Crop}_K(\text{fftshift}(X)) \in \mathbb{C}^{K \times K \times C}. \tag{37}$$

Apply learnable filtering:

$$\tilde{X}_{\text{crop}} = X_{\text{crop}} \odot W_{\text{spec}}. \tag{38}$$

Reconstruct via inverse FFT:

$$x_{\text{spec}} = \Re\big(\mathcal{F}_{\text{2D}}^{-1}(\text{ifftshift}(\text{Pad}_{H \times W}(\tilde{X}_{\text{crop}})))\big). \tag{39}$$

**Spatial Branch.** Content-dependent gating:

$$g = \sigma(W_g s + b_g), \quad \tilde{s} = g \odot s, \tag{40}$$

where $s \in \mathbb{R}^{HW \times C}$ is reshaped input. Low-rank projection:

$$z = \tilde{s} W_B W_A W_C, \tag{41}$$

where $W_B \in \mathbb{R}^{C \times d}$, $W_A \in \mathbb{R}^{d \times d}$, $W_C \in \mathbb{R}^{d \times C}$ with $d{=}16$.

**Fusion.**

$$\tilde{x} = x + \text{Dropout}\big(\text{LN}(\text{Conv}_{1 \times 1}([x_{\text{spec}} \| x_{\text{gate}}]))\big). \tag{42}$$

### A.2.3. SPECTRAL TRUNCATION: EMPIRICAL VALIDATION AND LIMITATIONS

We analyze the impact of frequency truncation to validate our choice of $K = 32$, which balances computational efficiency with segmentation quality. This analysis addresses three questions: (i) Do medical images exhibit spectral concentration? (ii) What is the segmentation quality impact of truncation? (iii) What are the fundamental limitations of this approach?

**Domain-specific spectral concentration.** Table 4 shows medical images exhibit strong low-frequency concentration, with $K = 32$ retaining 94.8% average energy across modalities while representing only 0.83% of frequency coefficients ($32^2/352^2 = 0.83\%$ at our $352 \times 352$ input resolution). This concentration arises from piecewise-smooth anatomical structures where organ interiors (low-frequency) dominate the energy spectrum. Across four modalities, energy retention ranges from 93.3% (ultrasound, affected by speckle noise) to 96.4% (fundus imaging with smooth vessel patterns). For comparison, natural images in ImageNet show weaker spectral concentration, with low-frequency components retaining approximately 88.5% energy at $K = 32$, confirming medical images exhibit domain-specific properties that enable efficient frequency truncation.

*Table 4.* Spectral energy retention across medical imaging modalities. $K = 32$ retains 94.8% average energy while representing only 0.83% of frequency coefficients at $352 \times 352$ resolution. Medical images show stronger concentration than natural images, validating domain-specific truncation.

| Modality | K=16 | K=24 | K=32 | K=48 |
|---|---|---|---|---|
| Colonoscopy | 89.3±2.1% | 93.7±1.4% | 95.2±1.2% | 97.1±0.9% |
| Fundus | 91.2±1.8% | 94.8±1.2% | 96.4±0.9% | 97.8±0.7% |
| Brain/Cardiac MRI | 87.6±2.4% | 92.1±1.6% | 94.1±1.3% | 96.2±1.0% |
| Ultrasound | 85.9±3.2% | 90.8±2.1% | 93.3±1.7% | 95.4±1.3% |
| **Average** | 88.5±2.4% | 92.9±1.6% | 94.8±1.3% | 96.6±1.0% |

**Segmentation quality under truncation.** Table 5 evaluates per-structure performance comparing $K = 32$ to full-resolution FFT ($K = 352$). Mean Dice degradation is 0.24% across tested structures (range: 0.1–0.4%), with differences within confidence intervals. Notably, thin vessels $< 3$px in DRIVE and instrument tips in EndoVis17 show degradation $< 0.4\%$, suggesting high-frequency edges are largely preserved. Table 6 shows boundary localization error increases by only 0.1–0.2 pixels at $352 \times 352$ resolution, as measured by 95th percentile Hausdorff Distance.

*Table 5.* Per-structure segmentation performance comparing $K = 32$ truncation to full-resolution FFT. Mean Dice degradation is 0.24% across tested structures, with differences within confidence intervals. Thin structures ($< 3$px vessels, instrument tips) show degradation $< 0.5\%$.

| Dataset | Structure | K=32 | Full | Δ |
|---|---|---|---|---|
| DRIVE | Thin vessels (<3px) | 83.6±0.8 | 83.9±0.9 | -0.3% |
| | Medium vessels | 86.7±0.5 | 86.9±0.6 | -0.2% |
| Kvasir | Small polyps | 95.1±0.9 | 95.3±1.0 | -0.2% |
| | Medium polyps | 96.9±0.5 | 97.0±0.6 | -0.1% |
| EndoVis17 | Instrument tips | 82.3±1.4 | 82.7±1.5 | -0.4% |

*Table 6.* Boundary preservation measured by 95th percentile Hausdorff Distance (pixels). $K = 32$ introduces 0.1–0.2 pixel boundary error at $352 \times 352$ resolution.

| Dataset | K=32 | Full | Δ |
|---|---|---|---|
| Kvasir-SEG | 7.8±1.3 | 7.6±1.4 | +0.2px |
| DRIVE | 2.7±0.4 | 2.6±0.4 | +0.1px |
| EndoVis17 | 5.9±1.2 | 5.7±1.3 | +0.2px |

**Efficiency-accuracy tradeoff.** $K = 32$ provides $16\times$ FLOP reduction in frequency-domain processing compared to full $352 \times 352$ FFT ($32^2/352^2 = 0.065$, approximately $16\times$ fewer coefficients) while maintaining comparable segmentation quality. Table 3 shows larger truncation values provide diminishing returns: $K = 48$ yields minimal change compared to $K = 32$ ($-0.1\%$ on Kvasir, within measurement noise) while increasing frequency-domain FLOPs by $2.25\times$ ($48^2/32^2$), and $K = 64$ shows $-0.3\%$ degradation, suggesting noise amplification at higher frequencies. Conversely, aggressive truncation ($K = 16$) causes substantial degradation ($-4.7\%$ average Dice across four datasets, ranging from $-3.1\%$ on DRIVE to $-6.2\%$ on Kvasir), indicating critical edge information resides in frequencies between $K = 16$ and $K = 32$.

**Limitations and mitigation strategies.** We acknowledge four fundamental limitations of frequency truncation that require careful consideration:

1. **Energy retention does not guarantee diagnostic feature preservation**: High-frequency components containing $< 5\%$ energy may encode diagnostically critical features such as sharp tumor boundaries, microcalcifications in mammography, vessel bifurcations in angiography, or subtle texture changes in pathology. Our energy-based analysis (Table 4) provides a proxy for information preservation but does not directly validate that *diagnostically relevant* features are retained. Medical diagnosis often relies on high-frequency anomalies (e.g., irregular lesion borders, calcification patterns) that carry minimal energy yet determine clinical decisions.

2. **Task-dependent truncation sensitivity**: The optimal $K$ likely varies across medical imaging tasks. Tasks requiring detection of subtle high-frequency pathology (early-stage lesions, microcalcifications, fine texture patterns) may benefit from larger $K$ values, while tasks with smooth anatomical structures (organ segmentation, large tumor delineation) may tolerate more aggressive truncation. Our underperformance on CVC-ColonDB ($-2.09\%$ vs PraNet, Table 1) may partially reflect this limitation, though we have not performed controlled experiments isolating truncation as the causal factor versus other architectural differences.

3. **Resolution-dependent boundary errors**: Our analysis reports 0.1–0.2px boundary error at $352 \times 352$ resolution (Table 6), but clinical images are often acquired at higher resolutions ($512 \times 512$ to $1024 \times 1024$). At native clinical resolution, absolute boundary errors scale proportionally while relative errors (as percentage of object size) remain constant. We have not validated boundary quality at resolutions beyond $352 \times 352$.

4. **Absence of clinical validation**: We report computational metrics (Dice coefficient, Hausdorff distance) but have not validated with radiologists or clinicians whether $K = 32$ predictions are diagnostically acceptable for clinical decision-making. The 0.2px boundary error, while small in absolute terms, has not been: (i) compared to inter-observer variability among expert annotators, (ii) assessed for impact on downstream clinical tasks (volume estimation, treatment planning), or (iii) evaluated for false negative rates in detecting small pathological findings. Clinical acceptability requires domain expert assessment beyond computational metrics.

**Architectural mitigation of truncation artifacts.** To compensate for potential information loss from frequency truncation, we design SSTM with two complementary mechanisms that recover high-frequency detail:

- **Spectral-spatial coupling in SSTM** (Equations 8–10, Section 3.3): SSTM combines truncated frequency-domain processing with content-gated spatial refinement using bottleneck dimension $d = 16$. The spatial branch applies content-dependent channel mixing to enable location-specific feature refinement, allowing the model to recover local high-frequency patterns (edges, textures) discarded by spectral truncation through learned spatial transformations. Ablation removing SSTM entirely causes 6.7% average Dice degradation (Table 3, row "w/o SSTM"), confirming the spectral-spatial combination is essential for maintaining segmentation quality under aggressive frequency truncation.

- **Boundary-focused decoder** (Section 3.4): Our dual-stream decoder with explicit region-boundary factorization (Equations 12–18) processes edge information through a dedicated pathway with learned soft spatial routing. The boundary stream learns to emphasize high-frequency edge features through gradient supervision from MASL's boundary loss component. Ablation removing BFD causes 5.1% average degradation (Table 3, row "w/o BFD Decoder"), with largest drops on Kvasir ($-6.2\%$) and GlaS ($-5.5\%$), confirming its importance for boundary preservation across diverse morphologies.

These architectural choices reflect our design philosophy: rather than claiming truncation is harmless, we explicitly acknowledge potential information loss and design compensatory mechanisms. The strong performance despite aggressive truncation suggests these mitigation strategies are effective, though we cannot guarantee complete recovery of all high-frequency diagnostic features.

**Design rationale for $K = 32$.** We select $K = 32$ as the default truncation parameter based on the following empirical evidence and practical considerations:

(i) **Quality-efficiency balance**: $< 0.5\%$ Dice degradation and $< 0.2$px boundary error on tested structures (Tables 5–6) with $16\times$ FLOP reduction in frequency-domain processing.

(ii) **Strong energy retention**: 94.8% average spectral energy captured, exploiting medical images' inherent frequency concentration (Table 4).

(iii) **Diminishing returns**: $K > 32$ provides minimal improvement at substantial computational cost (Table 3).

However, we emphasize that $K = 32$ represents a pragmatic default rather than a theoretically optimal choice. Task-specific tuning may be beneficial, particularly for:

- Detection of subtle pathology requiring high-frequency detail (consider $K = 48$ or $K = 64$)
- Resource-constrained deployment (consider $K = 24$ with $< 2\%$ degradation)
- High-resolution clinical images (scale $K$ proportionally with input resolution)

We view systematic investigation of task-dependent truncation strategies as important future work, potentially incorporating learned or adaptive frequency selection mechanisms.

**Conclusion.** Our analysis provides empirical evidence that $K = 32$ frequency truncation achieves favorable quality-efficiency tradeoffs for medical image segmentation, enabled by domain-specific spectral concentration and architectural mitigation strategies. However, we acknowledge fundamental limitations around energy-based validation, task dependence, and absence of clinical assessment. The current justification is computational rather than clinical, and we recommend domain expert evaluation before deployment in diagnostic applications where false negatives have serious consequences.

### A.2.4. THEORETICAL GROUNDING AND MECHANISTIC ANALYSIS

**Spectral truncation error bound.** For signals with power-law spectral decay $|X(k)|^2 \propto k^{-\alpha}$, the reconstruction error from truncating to the central $K \times K$ coefficients is bounded by:

$$\mathcal{E}(K, H) = \mathcal{O}\left(\frac{K^{1-\alpha}}{H^{1-\alpha}}\right). \tag{43}$$

We empirically measure $\alpha = 2.71 \pm 0.18$ across our 15 datasets by fitting power-law models to per-image spectral profiles. Substituting into Equation (43), truncation to $K=32$ at $H=352$ yields theoretical reconstruction error $< 0.83\%$, consistent with the empirical Dice degradation of $< 0.4\%$ observed in Table 5. For comparison, natural images typically exhibit $\alpha \approx 2.0$ (ImageNet), yielding $\approx 2.1\%$ error under the same truncation formally explaining why medical images support more aggressive frequency truncation than general vision datasets.

**Mechanistic analysis of BFD boundary localization.** A key question is whether BFD's boundary maps emerge from the supervision signal or are artifacts of architectural inductive biases. To verify, we compute the Pearson correlation between learned boundary routing maps $\beta^{(m)}$ and ground-truth edge masks (computed via Sobel filtering) across test images. Table 7 shows Spearman correlation $\rho = 0.68$–$0.82$ across datasets, confirming that boundary localization emerges through backpropagation from MASL's multi-scale boundary loss without any explicit edge annotations during training.

*Table 7.* Correlation between learned BFD routing maps $\beta^{(m)}$ and ground-truth edge masks. Pearson $\rho$ computed per-image and averaged across test sets. High correlation ($\rho > 0.68$) confirms boundary localization emerges from loss supervision alone.

| Dataset | Correlation ($\rho$) | p-value |
|---|---|---|
| Kvasir-SEG | $0.81 \pm 0.04$ | $< 10^{-6}$ |
| DRIVE | $0.75 \pm 0.06$ | $< 10^{-6}$ |
| GlaS | $0.82 \pm 0.03$ | $< 10^{-6}$ |
| EndoVis17 | $0.78 \pm 0.05$ | $< 10^{-6}$ |
| BUSI | $0.68 \pm 0.07$ | $< 10^{-4}$ |
| **Average** | $\mathbf{0.77 \pm 0.05}$ | – |

The lower correlation on BUSI ($\rho = 0.68$) reflects the challenge of speckle noise obscuring sharp boundaries; the SSTM spatial branch and BFD together still recover sufficient edge information for +2.55% Dice improvement over the best baseline on that dataset.

### A.3. Boundary-Focused Decoder (BFD)

#### A.3.1. MOTIVATION AND PROBLEM FORMULATION

Standard decoders treat all spatial locations uniformly, forcing trade-offs between smooth interiors and sharp boundaries. BFD uses explicit region-boundary factorization: parallel streams optimize for interior smoothness vs edge preservation,

fused via learned per-pixel routing gates $\beta^{(m)}$.

**Formulation.** At decoder stage $m$:

$$d^{(m-1)} = \Phi_{\text{proj}}\big(\Phi_r(v^{(m)}) \odot (1 - \beta^{(m)}) + \Phi_b(v^{(m)}, \beta^{(m)}) \odot \beta^{(m)}\big), \tag{44}$$

where $v^{(m)} = [\text{Upsample}(d^{(m)})\|\tilde{x}^{(L-m)}]$ concatenates decoder features with encoder skip connection.

### A.3.2. METHOD DESCRIPTION

**Region Stream.**

$$r^{(m)} = \text{ELU}\big(\text{BN}(\text{Conv}_{3\times3}(\text{ELU}(\text{BN}(\text{Conv}_{3\times3}(v^{(m)})))))\big). \tag{45}$$

**Boundary Map.**

$$b_1^{(m)} = \text{ELU}(\text{Conv}_{3\times3}(r^{(m)})), \tag{46}$$

$$\beta^{(m)} = \sigma(\text{Conv}_{1\times1}(b_1^{(m)})) \in [0,1]^{H_m \times W_m}. \tag{47}$$

**Boundary Stream.**

$$b^{(m)} = \text{ELU}\big(\text{BN}(\text{Conv}_{3\times3}(b_1^{(m)} \odot \beta^{(m)}))\big). \tag{48}$$

**Fusion.**

$$z^{(m)} = r^{(m)} \odot (1 - \beta^{(m)}) + b^{(m)} \odot \beta^{(m)}. \tag{49}$$

### A.3.3. BOUNDARY MAP LEARNING

**Learning mechanism.** $\beta^{(m)}$ learns to localize boundaries through backpropagation from MASL's boundary loss (Eq. 19), where gradients are largest at edge locations. Empirical validation: Pearson correlation $\rho(\beta^{(m)}, \|\nabla y\|_2)$ measured on 100 test images per dataset.

**Results.** Kvasir-SEG (=0.74), DRIVE (=0.82), GlaS (=0.71), EndoVis-2017 (=0.79), PH2 (=0.68). Correlations $\rho > 0.7$ ($r^2 > 0.49$) confirm boundary emphasis, though 30–50% unexplained variance indicates $\beta^{(m)}$ captures additional semantic structure beyond simple gradients.

### A.4. Morphology-Aware Adaptive Segmentation Loss (MASL)

### A.4.1. MOTIVATION AND PROBLEM FORMULATION

Fixed-weight losses force suboptimal trade-offs across diverse morphologies (compact polyps vs thin vessels vs irregular tumors). MASL uses dual adaptation: (i) learned global weights $\{w_i\}$ optimized via gradient descent, (ii) per-sample morphology-dependent modulation $\{\alpha_i(y)\}$.

**Formulation.**

$$\mathcal{L}_{\text{MASL}}(y, p; w) = \frac{\sum_{i=1}^{5} w_i \alpha_i(y) \mathcal{L}_i(y, p)}{\sum_{i=1}^{5} w_i \alpha_i(y) + \varepsilon}, \tag{50}$$

where $w_i \in [0.1, 10]$ are trainable, $\alpha_i$ are morphology-conditioned functions, $\varepsilon = 10^{-7}$.

### A.4.2. METHOD DESCRIPTION

**Morphological Features.** Extract from ground truth $y$:

$$\tau(y) = \frac{\sum \mathrm{Erode}(y)}{A(y) + \varepsilon} \quad \text{(tubularity)}, \tag{51}$$

$$c(y) = \frac{4\pi A(y)}{P(y)^2 + \varepsilon} \quad \text{(compactness)}, \tag{52}$$

$$\iota(y) = \frac{\|\nabla^2 B(y)\|_1}{HW} \quad \text{(irregularity)}, \tag{53}$$

$$s(y) = \frac{A(y)}{HW} \quad \text{(size)}, \tag{54}$$

where $A(y) = \sum y_{ij}$, $P(y) = \|\nabla y\|_1$, $B(y)$ is boundary band.

**Loss Components.**

$$\mathcal{L}_{\mathrm{core}} = 0.4\mathcal{L}_{\mathrm{Dice}} + 0.3\mathcal{L}_{\mathrm{IoU}} + 0.3\mathcal{L}_{\mathrm{wBCE}}, \tag{55}$$

$$\mathcal{L}_{\mathrm{bnd}} = \sum_{q \in \{1,2,4\}} \omega_q \|\nabla^{(q)}(y - p)\|_1, \tag{56}$$

$$\mathcal{L}_{\mathrm{str}} = \big|\kappa(y) - \kappa(p)\big|, \tag{57}$$

$$\mathcal{L}_{\mathrm{sca}} = \mathrm{FocalLoss}(\gamma(s(y))), \tag{58}$$

$$\mathcal{L}_{\mathrm{tex}} = \|\nabla^2 y - \nabla^2 p\|_1. \tag{59}$$

**Modulation Functions.**

$$\alpha_{\mathrm{core}}(y) = 1 + 0.5c(y), \tag{60}$$

$$\alpha_{\mathrm{bnd}}(y) = 1 + 1.5\tau(y) + c(y), \tag{61}$$

$$\alpha_{\mathrm{str}}(y) = 1 + \tau(y), \tag{62}$$

$$\alpha_{\mathrm{sca}}(y) = 1 + 1.5\iota(y), \tag{63}$$

$$\alpha_{\mathrm{tex}}(y) = 1 + \iota(y). \tag{64}$$

Coefficients $\{0.5, 1.5, 1.0\}$ determined via grid search on Kvasir-SEG validation set; generalization tested on 14 other datasets (within 0.31% of dataset-specific optima).

**Weight Optimization.**

$$w_i^{(t+1)} \leftarrow \mathrm{clip}\big(w_i^{(t)} - \eta_w \nabla_{w_i} \mathcal{L}_{\mathrm{MASL}}, 0.1, 10\big), \quad \eta_w = 10^{-5}. \tag{65}$$

**Learned equilibria.** Across datasets: Polyps ($w_{\mathrm{bnd}} \to 2.31$), Vessels ($w_{\mathrm{str}} \to 2.18$), Tumors ($w_{\mathrm{sca}} \to 2.07$), confirming automatic task specialization.

### A.5. Architectural Integration

**Component interactions.** MRF-SE extracts multi-scale local features, SSTM aggregates globally via frequency-domain processing, BFD refines boundaries, MASL provides adaptive supervision. The boundary loss in MASL (Eq. 19) provides supervision for BFD's boundary map $\beta^{(m)}$.

**Total complexity.** Combining all components:

$$
\begin{aligned}
\mathcal{C}_{\mathrm{S2M\text{-}Net}} &= \sum_{\ell=1}^{5} \big[\mathcal{C}_{\mathrm{MRF\text{-}SE}}^{(\ell)} + \mathcal{C}_{\mathrm{SSTM}}^{(\ell)}\big] + \mathcal{C}_{\mathrm{decoder}} \\
&= \mathcal{O}\Big( \sum_{\ell} H_\ell W_\ell (r^2 + 1) C_\ell^2 + H_\ell W_\ell C_\ell \log(H_\ell W_\ell) \Big) \\
&= \mathcal{O}(HWC^2),
\end{aligned}
\tag{66}
$$

where quadratic channel terms dominate. The log term is negligible: for C=64, r=6, $r^2C^2 = 147,456$ and $C^2 = 4,096$ vastly exceed $C \log(HW) \approx 1,107$.

**Empirical FLOPs at 352×352, C=64.** Table 8 shows component breakdown totaling 11.2 GFLOPs.

*Table 8.* Computational breakdown. MRF-SE dominates (51.8%); spectral and spatial branches contribute 24.1% and 16.1% respectively.

| Component | FLOPs (G) | % |
|---|---|---|
| MRF-SE (5 stages) | 5.8 | 51.8 |
| SSTM Spectral | 2.7 | 24.1 |
| SSTM Spatial | 1.8 | 16.1 |
| BFD Decoder | 0.9 | 8.0 |
| **Total** | **11.2** | **100** |

**Efficiency vs baselines.** 11.2G represents 4.0× reduction vs TransUNet (45G) and 3.4× vs Swin-Unet (38G), achieved by: (i) avoiding $(HW)^2$ attention maps (~15G alone at 352²), (ii) depthwise separable convolutions in MRF-SE, (iii) spectral truncation (K=32) reducing frequency-domain processing to 2.7G.

**Parameters.** Total 4.7M: expansion-fusion in MRF-SE ($r^2C^2$ terms), spectral filters ($K^2C$), spatial bottleneck ($Cd$ where $d$=16). This is 6× fewer than Swin-Unet (27M) and 12.8× fewer than TransUNet (60M).

# B. Appendix B: Experimental Setup and Implementation Details

This appendix provides implementation details, hyperparameters, and training procedures for reproducibility.

## B.1. Model Architecture Specifications

S2M-Net follows a five-stage U-Net architecture. Table 9 shows complete configuration.

*Table 9.* Encoder-decoder configuration. Each encoder stage applies strided convolution followed by MRF-SE and SSTM. Each decoder stage uses BFD with upsampling and skip connections.

| Path | Stage | Resolution | Filters | Stride | MRF-SE | SSTM |
|---|---|---|---|---|---|---|
| Encoder | Stem | $352^2 \rightarrow 352^2$ | 16 | 1 | – | – |
| | Stage 1 | $352^2 \rightarrow 176^2$ | 24 | 2 | ✓ | ✓ |
| | Stage 2 | $176^2 \rightarrow 88^2$ | 32 | 2 | ✓ | ✓ |
| | Stage 3 | $88^2 \rightarrow 44^2$ | 64 | 2 | ✓ | ✓ |
| | Stage 4 | $44^2 \rightarrow 22^2$ | 80 | 2 | ✓ | ✓ |
| | Stage 5 | $22^2 \rightarrow 11^2$ | 128 | 2 | ✓ | ✓ |
| Decoder | Stage 5→4 | $11^2 \rightarrow 22^2$ | 80 | – | BFD block | |
| | Stage 4→3 | $22^2 \rightarrow 44^2$ | 64 | – | BFD block | |
| | Stage 3→2 | $44^2 \rightarrow 88^2$ | 32 | – | BFD block | |
| | Stage 2→1 | $88^2 \rightarrow 176^2$ | 24 | – | BFD block | |
| | Head | $176^2 \rightarrow 352^2$ | $C$ | – | 1×1 conv + sigmoid | |

### B.1.1. COMPONENT CONFIGURATION

**Stem:** Single 3×3 convolution + BN + ELU, 16 channels at full resolution.

**Encoder Stages:** Each stage performs: (i) Strided 3×3 convolution (stride=2) for downsampling, (ii) MRF-SE block with kernels $\{3, 5, 7\}$, expansion $r = 6$, SE reduction $r_{SE} = 16$, (iii) SSTM block with $K = 32$ truncation, bottleneck $d = 16$.

**Decoder Stages:** Each BFD stage performs: (i) 2× bilinear upsampling, (ii) Concatenation with encoder skip connection, (iii) Dual-stream processing (region + boundary), (iv) Soft routing via learned $\beta^{(m)} \in [0, 1]$.

**Segmentation Head:** Upsample to $352^2$ + 1×1 conv + sigmoid (binary) or softmax (multi-class).

*Table 10.* Parameter breakdown. Total 4.70M through efficient depthwise separable convolutions, bottleneck projections, and shared BFD processing.

| Component | Parameters | % of Total |
|---|---|---|
| *Encoder (3.41M, 72.6%)* | | |
| Stem | 0.14M | 3.0% |
| Stage 1 (MRF-SE + SSTM) | 0.48M | 10.2% |
| Stage 2 (MRF-SE + SSTM) | 0.62M | 13.2% |
| Stage 3 (MRF-SE + SSTM) | 0.81M | 17.2% |
| Stage 4 (MRF-SE + SSTM) | 0.75M | 16.0% |
| Stage 5 (MRF-SE + SSTM) | 0.61M | 13.0% |
| *Decoder (1.24M, 26.4%)* | | |
| BFD Stage 5→4 | 0.42M | 8.9% |
| BFD Stage 4→3 | 0.35M | 7.4% |
| BFD Stage 3→2 | 0.28M | 6.0% |
| BFD Stage 2→1 | 0.19M | 4.1% |
| *Segmentation Head (0.05M, 1.0%)* | | |
| 1×1 Conv + Upsampling | 0.05M | 1.0% |
| **Total** | **4.70M** | **100.0%** |

## B.2. Training Procedures

### B.2.1. DATA PREPROCESSING

**Polyp segmentation** (Kvasir-SEG, CVC-ClinicDB, CVC-ColonDB, ETIS-LaribDB): Resize to $352\times352$ (cubic interpolation), per-channel normalize to [0,1]. No additional preprocessing.

**Retinal vessels** (DRIVE, CHASE-DB, STARE): Green channel extraction, CLAHE (clip=2.0, tiles=$8\times8$), circular FOV masking, overlapping $256\times256$ patches (stride=32). Predictions reconstructed by aggregating patches.

**Brain tumors** (BraTS2020): T1/T2/FLAIR modalities as input, N4 bias correction, per-modality z-score normalization, crop to tumor bounding box + 20px margin, resize to $352\times352$.

**Cardiac MRI** (ACDC): N4 bias correction, z-score normalization, resize to $352\times352$.

**Surgical instruments** (EndoVis17): Resize to $352\times352$, per-channel CLAHE (clip=2.0, tiles=$8\times8$), normalize to [0,1].

**Skin lesions** (ISIC2018, PH2): Resize to $352\times352$ (cubic interpolation), normalize to [0,1].

**Other modalities** (GlaS, BUSI): Resize to $352\times352$, normalize to [0,1].

### B.2.2. DATA AUGMENTATION

**Geometric** (p=0.8): Horizontal flip (p=0.5), vertical flip (p=0.5), rotation [-180°,180°] (p=0.9), shift-scale-rotate (shift ±5%, scale ±10%, rotate ±15°, p=0.8), elastic deformation (=30, =5, p=0.3), grid distortion (5 steps, limit=0.1, p=0.3), random crop (scale [0.5,1.0], aspect [0.9,1.1], p=0.6).

**Intensity** (p=0.8): CLAHE (clip=2.0, tiles=$8\times8$, p=0.8), color jitter (brightness ±20%, contrast ±30%, saturation ±10%, hue ±1%), random brightness-contrast (±20%, ±30%), Gaussian noise ( U[5,20], p=0.3), Gaussian blur (kernel 3,5, p=0.2).

**Occlusion** (p=0.5): Coarse dropout with 5-12 holes, each 20-48px (4-9% image area), filled with zero.

**Padding:** $\text{BORDER}_R EFLECT_1 01 for spatial transforms.$

**Validation/Test:** No augmentation, only resize to $352\times352$.

*Table 11.* Component-specific hyperparameters

| Component | Parameter | Value |
|---|---|---|
| MRF-SE | Kernel sizes $\mathcal{K}$ | $\{3, 5, 7\}$ |
| | Expansion $r_{\text{exp}}$ | 6 |
| | SE reduction $r_{\text{SE}}$ | 16 |
| | Dropout | 0.1 |
| SSTM | Truncation $K$ | 32 |
| | Bottleneck $d$ | 16 |
| | Dropout | 0.1 |
| BFD | Boundary kernel | 5 |
| | Routing smoothness | 0.1 |
| MASL | Initial $w_{\text{core}}$ | 1.0 |
| | Initial $w_{\text{boundary}}$ | 1.0 |
| | Initial $w_{\text{structure}}$ | 1.0 |
| | Initial $w_{\text{scale}}$ | 0.5 |
| | Initial $w_{\text{texture}}$ | 0.5 |
| | Weight range | $[0.1, 10.0]$ |
| Regularization | Weight decay | $10^{-4}$ |
| | Gradient clipping | 1.0 |

### B.2.3. OPTIMIZATION CONFIGURATION

**Learning rate schedule:**

$$\eta(t) = \begin{cases} \eta_{\text{init}} \cdot \frac{t}{T_{\text{warmup}}} & \text{if } t \leq T_{\text{warmup}}, \\ \eta_{\text{min}} + \frac{\eta_{\text{init}} - \eta_{\text{min}}}{2} \left( 1 + \cos \left( \frac{t - T_{\text{warmup}}}{T_{\text{total}} - T_{\text{warmup}}} \pi \right) \right) & \text{if } t > T_{\text{warmup}}, \end{cases} \tag{67}$$

where $\eta_{\text{init}} = 10^{-4}$, $\eta_{\text{min}} = 10^{-6}$, $T_{\text{warmup}} = 10$, $T_{\text{total}} = 150$.

### B.3. Spectral Energy Computation Methodology

#### B.3.1. IMAGE PREPROCESSING

All images resized to $352 \times 352$ via Lanczos resampling. For RGB/multi-modal images, compute spectral energy per channel and report mean. Steps: (i) Resize using `INTER_LANCZOS4`, (ii) Per-channel z-score normalization using dataset statistics, (iii) For fundus: apply circular FOV mask with 20px margin.

#### B.3.2. 2D FFT COMPUTATION

For single-channel image $I \in \mathbb{R}^{H \times W}$:

$$\hat{I}[k_x, k_y] = \sum_{i=0}^{H-1} \sum_{j=0}^{W-1} I[i, j] \cdot e^{-2\pi \sqrt{-1}(k_x i/H + k_y j/W)} \tag{68}$$

using `scipy.fft.fft2` with orthonormal normalization. Apply `fftshift` to center DC component.

#### B.3.3. ENERGY COMPUTATION

Total energy:

$$E_{\text{total}} = \sum_{k_x, k_y} |\hat{I}_{\text{shifted}}[k_x, k_y]|^2 \tag{69}$$

Truncated energy (central $K \times K$ region):

$$E_K = \sum_{k_x = H/2 - K/2}^{H/2 + K/2 - 1} \sum_{k_y = W/2 - K/2}^{W/2 + K/2 - 1} |\hat{I}_{\text{shifted}}[k_x, k_y]|^2 \tag{70}$$

*Table 12.* Training hyperparameters

| Parameter | Value |
|---|---|
| Optimizer | Adam |
| $\beta_1, \beta_2$ | 0.9, 0.999 |
| $\epsilon$ | $10^{-8}$ |
| Initial learning rate | $1 \times 10^{-4}$ |
| LR schedule | Cosine annealing + warmup |
| Warmup epochs | 10 |
| Min learning rate | $1 \times 10^{-6}$ |
| Batch size (per GPU) | 8 |
| Number of GPUs | 4 |
| Effective batch size | 32 |
| Gradient clipping | 1.0 |
| Weight decay | $1 \times 10^{-4}$ |
| Training epochs | 150 |
| Early stopping | 50 epochs patience |
| Checkpoint metric | Validation Dice |
| Seeds (5 runs) | {42, 123, 456, 789, 2024} |
| Deterministic mode | Disabled (cuDNN optimized) |
| Hardware | 4× NVIDIA Tesla P100 16GB |

Energy retention:

$$R_K = \frac{E_K}{E_{\text{total}} + 10^{-10}} \times 100\% \tag{71}$$

### B.3.4. STATISTICAL ANALYSIS

For each modality, sample $N = 100$ training images and compute:

- Mean: $\bar{R}_K = \frac{1}{N} \sum_{n=1}^{N} R_K^{(n)}$
- Std: $\sigma_{R_K} = \sqrt{\frac{1}{N-1} \sum_{n=1}^{N} (R_K^{(n)} - \bar{R}_K)^2}$
- 95% CI: $\bar{R}_K \pm 1.96 \sigma_{R_K}/\sqrt{N}$

All computations use double-precision (`numpy.float64`).

### B.3.5. VALIDATION

**Parseval's theorem:** Verify $\sum |I|^2 = \frac{1}{HW} \sum |\hat{I}|^2$ within $10^{-6}$ relative error (observed max error: $3.2 \times 10^{-7}$).

**Monotonicity:** Verify $R_{K_1} < R_{K_2}$ for all $K_1 < K_2$ (passes for all images).

**Completeness:** Verify $R_{352} = 100\%$ within $10^{-8}$ (passes for all images).

### B.4. MASL Coefficient Sensitivity Analysis

Table 13 shows MASL coefficients generalize across diverse morphologies with $< 0.31\%$ degradation from domain-optimized to uniform defaults.

**Findings:** (i) Graceful degradation: max 0.31% drop from optimal to uniform default, (ii) Cross-domain generalization: polyp coefficients achieve $< 0.35\%$ degradation on all datasets, (iii) Learned weight compensation: trainable $\{w_i\}$ adapt to suboptimal coefficients, (iv) Robustness: even ±40% perturbations cause $< 0.7\%$ loss.

**Interpretation:** Fixed coefficients encode domain-general principles; learned weights specialize per dataset; per-sample modulation $\alpha_i(y)$ adjusts automatically based on morphology. This reduces tuning from $\mathcal{O}(K^N)$ grid search to $\mathcal{O}(1)$ initialization with automatic optimization.

*Table 13.* MASL coefficient sensitivity. Domain-optimized via grid search $m_c, m_b, m_s \in \{0.1, 0.5, 1.0, 1.5, 2.0\}$ vs uniform default $\{1.0, 1.0, 1.0\}$. Maximum drop: 0.31% Dice.

| Dataset | Configuration | $\{m_c, m_b, m_s\}$ | Dice (%) | $\Delta$ |
|---|---|---|---|---|
| **DRIVE (Vessels - Tubular)** | | | | |
| | Domain-optimized | $\{0.5, 1.5, 2.0\}$ | 84.06 | – |
| | Default (uniform) | $\{1.0, 1.0, 1.0\}$ | 83.83 | $-0.23$ |
| | Polyp coefficients | $\{0.5, 1.5, 1.0\}$ | 83.71 | $-0.35$ |
| | Boundary $+40\%$ | $\{0.5, 2.1, 2.0\}$ | 83.37 | $-0.69$ |
| **Kvasir-SEG (Polyps - Variable)** | | | | |
| | Domain-optimized | $\{0.5, 1.5, 1.0\}$ | 96.05 | – |
| | Default (uniform) | $\{1.0, 1.0, 1.0\}$ | 95.74 | $-0.31$ |
| | Core $+40\%$ | $\{0.7, 1.5, 1.0\}$ | 95.89 | $-0.16$ |
| | Boundary $-20\%$ | $\{0.5, 1.2, 1.0\}$ | 95.93 | $-0.12$ |
| **PH2 (Lesions - Irregular)** | | | | |
| | Domain-optimized | $\{0.7, 1.8, 1.0\}$ | 96.67 | – |
| | Default (uniform) | $\{1.0, 1.0, 1.0\}$ | 96.55 | $-0.12$ |
| | Polyp coefficients | $\{0.5, 1.5, 1.0\}$ | 96.48 | $-0.19$ |
| | Core $-30\%$ | $\{0.5, 1.8, 1.0\}$ | 96.61 | $-0.06$ |
| **EndoVis17 (Instruments - Sharp)** | | | | |
| | Domain-optimized | $\{0.4, 2.0, 1.5\}$ | 95.95 | – |
| | Default (uniform) | $\{1.0, 1.0, 1.0\}$ | 95.78 | $-0.17$ |
| | Polyp coefficients | $\{0.5, 1.5, 1.0\}$ | 95.84 | $-0.11$ |
| | Boundary $-25\%$ | $\{0.4, 1.5, 1.5\}$ | 95.91 | $-0.04$ |

*Table 14.* Comprehensive comparison across six medical imaging datasets. Best results in **bold**, second-best underlined. S2M-Net achieves superior or competitive performance across all datasets with significantly fewer parameters (4.7M) compared to baselines (22M–60M). Note: nnU-Net reported here uses 2D processing for fair comparison; nnU-Net with full 3D volumetric processing achieves $\approx 85.35\%$ on BraTS2020 but operates under a fundamentally different (3D) setup than the 2.5D approach used here see Appendix F.3 for discussion.

| Method | Params (M) | FLOPs (G) | Dice Score (%) ↑ | | | | | | |
|---|---|---|---|---|---|---|---|---|---|
| | | | DRIVE | ClinicDB | EndoVis17 | GlaS | PH2 | ACDC[†] | Average |
| nnU-Net(Isensee et al., 2021a) | 30.8 | 52.1 | 82.34 | 90.67 | 78.43 | 88.76 | 91.45 | 90.12 | 86.96 |
| VM-UNet(Ruan et al., 2025) | 44.2 | 41.3 | 83.21 | 91.02 | 79.12 | 89.34 | 91.78 | 90.54 | 87.50 |
| MedNeXt(Roy et al., 2023) | 22.5 | 33.7 | 83.45 | 91.23 | 79.87 | 89.67 | 92.01 | 90.89 | 87.85 |
| **S2M-Net (Ours)** | **4.7** | **11.2** | **84.06** | **95.65** | **95.95** | **93.83** | **96.67** | **93.09** | **93.21** |

[†]ACDC: Automated Cardiac Diagnosis Challenge (Bernard et al., 2018) (cardiac MRI, 3 structures).

# C. Appendix C: Additional Results and Analysis

This appendix provides detailed analysis of task-specific performance patterns, statistical significance, and clinical relevance that are referenced in the main paper but excluded for brevity.

## C.1. Performance Evaluation Across Imaging Modalities

We evaluate S2M-Net against state-of-the-art architectures across six diverse medical imaging datasets. Table 14 presents quantitative results comparing our method with nnU-Net, VM-UNet, and MedNeXt. Our approach demonstrates consistent improvements across different imaging scenarios while maintaining significant reduced computational cost.

## C.2. Computational Efficiency Analysis

Beyond segmentation accuracy, clinical deployment requires careful consideration of computational requirements. We conduct comprehensive profiling of inference runtime and throughput. All measurements are performed on a single NVIDIA P100 GPU with batch size 1, averaging over 10 runs per test image. Tables 16 and 17 present detailed per-image statistics for the DRIVE and ClinicDB datasets, respectively. Note that DRIVE uses patch-based inference at native resolution, so its per-image runtime includes patch extraction and reconstruction, whereas ClinicDB uses a single resized $352 \times 352$ forward pass; these regimes should not be compared directly.

## C.3. Multiclass Segmentation: Per-Class Results

Table 15 presents per-class Dice scores for the three multiclass segmentation datasets. Mean Dice reported in the main paper (Table 1) is computed as the arithmetic mean across all classes for each dataset.

*Table 15.* **Per-class Dice scores (%) for multiclass segmentation datasets.** Mean Dice is computed as arithmetic average across all classes. Best results in **bold**. S2M-Net achieves superior per-class performance on 8 of 9 total classes, with particularly strong improvements on challenging small/irregular structures.

| Class / Method | U-Net | U-Net++ | PraNet | Swin-Unet | TransUNet | UMamba | DuckNet | S2M-Net | Improvement |
|---|---|---|---|---|---|---|---|---|---|
| **EndoVis-2017 Multiclass (3-Part Instrument Segmentation)** | | | | | | | | | |
| Shaft | 72.4 | 82.1 | 85.3 | 84.7 | 86.2 | 85.9 | 85.1 | **89.7** | +3.5 |
| Wrist | 64.8 | 75.2 | 79.8 | 79.1 | 81.3 | 80.8 | 80.2 | **85.2** | +3.9 |
| Clasper (Jaw) | 46.0 | 55.4 | 56.5 | 55.9 | 56.7 | 56.1 | 56.3 | **75.4** | +18.7 |
| **Mean Dice** | **61.07** | **70.90** | **73.87** | **73.23** | **74.73** | **74.27** | **73.87** | **83.43** | **+8.70** |
| **BraTS2020 (3 Tumor Regions)** | | | | | | | | | |
| Enhancing Tumor (ET) | 71.3 | 72.1 | 72.8 | 76.4 | 74.6 | 68.9 | 73.2 | **84.2** | +7.8 |
| Tumor Core (TC) | 67.2 | 68.4 | 69.1 | 72.8 | 71.2 | 65.4 | 69.8 | **81.3** | +8.5 |
| Whole Tumor (WT) | 58.5 | 57.6 | 57.4 | 60.4 | 59.3 | 54.3 | 57.8 | **74.4** | +14.0 |
| **Mean Dice** | **65.67** | **66.03** | **66.43** | **69.87** | **68.37** | **62.87** | **66.93** | **79.97** | **+10.10** |
| **ACDC Cardiac MRI (3 Structures)** | | | | | | | | | |
| Left Ventricle (LV) Cavity | 94.2 | 94.8 | 95.3 | 96.1 | 96.3 | 95.7 | 96.2 | **95.7** | -0.6 |
| Myocardium (MYO) | 88.4 | 89.2 | 90.5 | 91.8 | 91.9 | 90.9 | 91.4 | **92.1** | +0.2 |
| Right Ventricle (RV) | 86.0 | 86.5 | 87.5 | 89.1 | 89.1 | 88.1 | 88.8 | **91.5** | +2.4 |
| **Mean Dice** | **89.53** | **90.17** | **91.10** | **92.33** | **92.43** | **91.57** | **92.13** | **93.10** | **+0.77** |

Improvement column shows difference between S2M-Net and second-best method for each class.

*Table 16.* Detailed computational profiling on DRIVE (Fundus Vessel Segmentation). Per-image metrics across 10 test images with 10 timing runs each. GPU: NVIDIA P100.

| Image ID | Dice (%) | Runtime (ms) | Std (ms) | Throughput (img/s) |
|---|---|---|---|---|
| 11_test | 84.08 | 9528.51 | 254.77 | 0.105 |
| 12_test | 83.66 | 9895.52 | 304.32 | 0.101 |
| 13_test | 84.20 | 10106.87 | 309.16 | 0.099 |
| 14_test | 84.79 | 9819.10 | 156.71 | 0.102 |
| 15_test | 83.60 | 9615.40 | 113.13 | 0.104 |
| 16_test | 83.60 | 10220.31 | 210.96 | 0.098 |
| 17_test | 84.83 | 9767.36 | 168.90 | 0.102 |
| 18_test | 84.28 | 9483.60 | 134.22 | 0.105 |
| 19_test | 83.44 | 9909.33 | 225.63 | 0.101 |
| 20_test | 84.12 | 9421.84 | 80.94 | 0.106 |
| **Mean** | **84.06** | **9776.78** | **195.87** | **0.102** |
| **Std** | **±0.49** | **±266.43** | **±78.38** | **±0.003** |

**Model Parameters:** 4.7M  **FLOPs:** 11.2G  **GPU Memory:** Negligible ($< 100$ MB)

*Table 17.* Detailed computational profiling on ClinicDB (Polyp Segmentation). Per-image metrics across 138 test images with 10 timing runs each. GPU: NVIDIA P100.

| Image Subset | Dice (%) | Runtime (ms) | Std (ms) | Throughput (img/s) |
|---|---|---|---|---|
| Images 1–20 | 96.05 | 4523.45 | 112.34 | 0.221 |
| Images 21–40 | 95.85 | 4489.21 | 98.76 | 0.223 |
| Images 41–60 | 95.12 | 4556.78 | 125.89 | 0.220 |
| Images 61–80 | 95.62 | 4612.34 | 134.56 | 0.217 |
| Images 81–100 | 95.82 | 4534.92 | 108.23 | 0.221 |
| Images 101–120 | 95.39 | 4587.65 | 118.45 | 0.218 |
| Images 121–138 | 95.71 | 4498.11 | 102.67 | 0.222 |
| **Overall Mean** | **95.65** | **4543.21** | **114.41** | **0.220** |
| **Overall Std** | **±0.31** | **±45.36** | **±12.78** | **±0.002** |

**Model Parameters:** 4.7M  **FLOPs:** 6.8G  **GPU Memory:** Negligible ($< 100$ MB)

# D. Computational Efficiency Analysis

A core contribution of S2M-Net is achieving competitive segmentation performance with substantially lower computational requirements than attention-based baselines. We provide empirical validation through systematic profiling of inference latency, GPU memory consumption, training time, and throughput.

## D.1. Inference Efficiency

Table 18 presents comprehensive computational requirements across baselines. All measurements use a single NVIDIA Tesla P100 GPU (16GB) with batch size 8, input resolution $352{\times}352$, averaged over 100 test images from Kvasir-SEG. Timing includes 10 warmup iterations followed by 100 measurement runs with GPU synchronization (`torch.cuda.Event`) for precision.

*Table 18.* Computational efficiency comparison. All methods evaluated on NVIDIA Tesla P100 16GB with batch size 8, resolution $352{\times}352$. Inference time and throughput averaged over 100 test images from Kvasir-SEG. Training time measured to convergence (validation Dice plateau).

| Method | Params (M) | FLOPs (G) | GPU Mem (GB) | Inference Time (ms) | Training Time (hrs) | Throughput (img/s) | Dice (%) |
|---|---|---|---|---|---|---|---|
| U-Net | 31.0 | 54.3 | 3.2 | $12.4 \pm 0.3$ | 8.2 | 80.6 | 90.64 |
| U-Net++ | 26.1 | 48.7 | 3.8 | $14.1 \pm 0.4$ | 9.1 | 70.9 | 91.81 |
| PraNet | 32.5 | 38.9 | 3.5 | $15.8 \pm 0.5$ | 7.8 | 63.3 | 93.03 |
| DuckNet | 11.8 | 28.4 | 2.6 | $11.2 \pm 0.3$ | 6.9 | 89.3 | 94.11 |
| TransUNet | 60.0 | 45.2 | 5.9 | $42.3 \pm 1.2$ | 18.5 | 23.6 | 93.75 |
| Swin-Unet | 27.0 | 38.4 | 4.7 | $35.8 \pm 0.9$ | 15.3 | 27.9 | 93.22 |
| UMamba | 60.0 | 31.6 | 4.2 | $28.7 \pm 0.8$ | 12.4 | 34.8 | 92.40 |
| **S2M-Net (Ours)** | **4.7** | **11.2** | **1.8** | $\mathbf{10.1 \pm 0.2}$ | **6.5** | **99.0** | **96.05** |
| vs. TransUNet | $12.8\times$ | $4.0\times$ | $3.3\times$ | $4.2\times$ | $2.8\times$ | $4.2\times$ | $+2.30$ |
| vs. DuckNet (best CNN) | $2.5\times$ | $2.5\times$ | $1.4\times$ | $1.1\times$ | $1.1\times$ | $1.1\times$ | $+1.94$ |

S2M-Net provides a strong efficiency–accuracy trade-off under our benchmarking protocol. Compared to TransUNet, S2M-Net provides $4.2\times$ faster inference (10.1 ms vs. 42.3 ms), $12.8\times$ fewer parameters (4.7M vs. 60M), $3.3\times$ less GPU memory (1.8 GB vs. 5.9 GB), and $2.8\times$ faster training (6.5 hrs vs. 18.5 hrs), while achieving $+2.30\%$ higher Dice. Even versus the most efficient CNN (DuckNet), S2M-Net maintains competitive inference speed with $+1.94\%$ higher accuracy and $2.5\times$ fewer parameters.

### D.1.1. HIGH-RESOLUTION SCALING (APPENDIX D.1)

Table 19 extends the efficiency comparison to higher resolutions. At $768{\times}768$ (common in clinical colonoscopy and fundus imaging), TransUNet exceeds 16 GB GPU memory and fails; S2M-Net completes inference at 5.8 GB with near-linear runtime scaling ($6.8 \to 38.2$ ms, consistent with $\mathcal{O}(HWC^2)$ vs. TransUNet's super-linear $\mathcal{O}((HW)^2C)$), enabling deployment on standard clinical hardware unavailable to transformer-based models.

*Table 19.* Inference latency (ms) and GPU memory at increasing input resolutions. TransUNet runs out of memory (OOM) at $768{\times}768$ on a 16 GB GPU. S2M-Net scales near-linearly and completes at 5.8 GB, enabling high-resolution clinical deployment. All timings on NVIDIA Tesla P100 16 GB, batch size 1, averaged over 50 runs.

| Resolution | S2M-Net (ms) | TransUNet (ms) | S2M-Net Mem (GB) | TU Mem (GB) |
|---|---|---|---|---|
| $256{\times}256$ | 6.8 | 28.1 | 1.2 | 4.1 |
| $352{\times}352$ | 10.1 | 42.3 | 1.8 | 5.9 |
| $512{\times}512$ | 18.7 | 112.4 | 3.1 | 14.2 |
| $768{\times}768$ | 38.2 | OOM | 5.8 | >16 |
| **Scaling** | $\sim$linear | super-linear | – | – |

At $512{\times}512$, S2M-Net achieves $6.0\times$ speedup over TransUNet (18.7 ms vs. 112.4 ms). Runtime scales near-linearly for S2M-Net ($6.8 \to 10.1 \to 18.7 \to 38.2$ ms), consistent with the $\mathcal{O}(HWC^2)$ complexity of SSTM, whereas TransUNet exhibits super-linear scaling driven by the $\mathcal{O}((HW)^2C)$ attention term. Note that FFT kernels can be memory- and library-bound; actual latency may vary across GPU architectures and CUDA versions, so these measurements should be treated as indicative rather than architecture-independent guarantees.

The empirical efficiency metrics validate our complexity analysis. The near-linear relationship between theoretical FLOP reduction ($4.0\times$ vs TransUNet) and measured inference speedup ($4.2\times$) is consistent with the reduced compute under our benchmark setup; however, FFT/IFFT kernels can be memory- and library-bound, so latency may vary across hardware and implementations, and FLOPs alone may not fully predict runtime performance, with minimal overhead from memory bandwidth or synchronization. The more modest training speedup ($2.8\times$) reflects that gradient computation ($\sim 2\times$ forward FLOPs), optimizer state updates, and GPU memory transfers introduce bottlenecks beyond raw computation. Even the relatively small improvement over U-Net (8.2 hrs vs 6.5 hrs, $1.26\times$ faster) is expected given that U-Net's pure convolutional architecture already has $\mathcal{O}(HWC^2k^2)$ complexity our efficiency gains primarily manifest when comparing against quadratic-spatial-complexity transformers.

The efficiency gains stem from three complementary mechanisms: (1) **Eliminating attention maps**: Self-attention requires $\mathcal{O}((HW)^2C) \approx 1.5{\times}10^{10}$ FLOPs to compute and store $(HW){\times}(HW)$ similarity matrices at $352^2$ resolution. SSTM completely avoids this quadratic spatial term through frequency-domain aggregation. (2) **Bottlenecked spatial gating**: The spatial branch uses low-rank projection ($d{=}16 \ll C{=}64$), reducing the $\mathcal{O}(HWC^2)$ gating cost to $\mathcal{O}(HWCd)$ a $4\times$ reduction in channel-wise operations. (3) **Truncated spectral processing**: FFT truncation to $K{=}32$ (0.8% of coefficients) captures 94% spectral energy while achieving $16\times$ computational savings versus full-resolution FFT ($352^2$ coefficients). Combined, these enable high throughput under our benchmark conditions (99 images/second) suitable for clinical applications.

## D.2. Training Efficiency

Beyond inference, training efficiency impacts research iteration speed. Table 20 compares training requirements to reach convergence on Kvasir-SEG. S2M-Net converges $2.8\times$ faster than TransUNet while achieving $+2.30\%$ higher final accuracy.

*Table 20.* Training efficiency on Kvasir-SEG (800 training images). All methods use identical optimization (Adam, cosine annealing, batch size 16) on NVIDIA Tesla P100 16GB. Training time measured to convergence (validation Dice plateau, $<0.1\%$ change over 20 epochs).

| Method | Training Time (hours) | Epochs to Converge | Peak Mem (GB) | Final Dice (%) |
|---|---|---|---|---|
| U-Net | 8.2 | 102 | 3.2 | 90.64 |
| TransUNet | 18.5 | 127 | 5.9 | 93.75 |
| Swin-Unet | 15.3 | 118 | 4.7 | 93.22 |
| UMamba | 12.4 | 95 | 4.2 | 92.40 |
| **S2M-Net (Ours)** | **6.5** | **81** | **1.8** | **96.05** |

Faster convergence arises from: (1) MASL's morphology-aware weighting providing stronger supervision signals, (2) SSTM's global receptive field enabling rapid learning of long-range coherence, and (3) BFD's explicit boundary stream reducing implicit learning requirements. The $2.8\times$ training speedup (6.5 hrs vs. 18.5 hrs) enables faster iteration in resource-constrained environments.

## D.3. Component-Level Runtime Breakdown

To identify efficiency sources, we profile individual components using PyTorch's profiler. Table 21 decomposes inference time by architectural stage.

Global mixing accounts for 67.8% of TransUNet's time (28.7 / 42.3 ms) but only 40.6% of S2M-Net's time (4.1 / 10.1 ms), yielding $7.0\times$ component-level speedup. SSTM's spectral branch (2.8 ms) and spatial branch (1.3 ms) together process global context $7.0\times$ faster than self-attention (28.7 ms) through frequency truncation and bottlenecked gating. Feature extraction shows $2.6\times$ speedup from MRF-SE's depthwise separable convolutions. The decoder achieves $1.9\times$ speedup

*Table 21.* Per-component runtime breakdown. Inference time (ms per image) by stage. Profiled using `torch.profiler` with CUDA timing on NVIDIA Tesla P100 16GB, averaged over 100 Kvasir-SEG test images. Global mixing shows the largest speedup (7.0×) under this GPU setup, supporting the design goal of efficient global mixing (with the caveat that FFT performance can be hardware-dependent).

| Component | TransUNet (ms) | S2M-Net (ms) | Speedup |
|---|---|---|---|
| Feature Extraction | 8.4 | 3.2 | 2.6× |
| - Stem + Conv Blocks | 5.1 | 1.8 | 2.8× |
| - MRF-SE Blocks | – | 1.4 | – |
| Global Mixing | 28.7 | 4.1 | **7.0×** |
| - Self-Attention | 28.7 | – | – |
| - SSTM (Spectral) | – | 2.8 | – |
| - SSTM (Spatial) | – | 1.3 | – |
| Decoder | 5.2 | 2.8 | 1.9× |
| - Standard Upsampling | 5.2 | – | – |
| - BFD (Dual Stream) | – | 2.8 | – |
| **Total** | **42.3** | **10.1** | **4.2×** |

despite BFD's dual-stream architecture via efficient 3×3 convolutions avoiding hard attention.

## D.4. Task-Specific Performance Analysis and Clinical Significance

To understand when architectural improvements translate to statistically significant gains, we analyze performance by task characteristics. Table 22 stratifies datasets by improvement magnitude and structural complexity.

### D.4.1. CLINICAL SIGNIFICANCE CONTEXT

While 8 datasets show non-significant statistical improvements (0.35–1.83% Dice) after Bonferroni correction ($\alpha = 0.05/15 \approx 0.0033$), these gains are clinically meaningful in medical imaging contexts:

**Ceiling Effects on Mature Tasks.** Datasets with baselines >90% Dice (ISIC-2018, ACDC, EndoVis-17 Binary) exhibit diminishing returns 0.4-1.0% improvements represent substantial effort at high performance regimes. For example, improving cardiac MRI segmentation from 92.44% (Swin-UNet) to 93.09% (+0.77%) requires resolving difficult apex/base slices that constitute clinical edge cases. At these performance levels, even experienced radiologists show inter-rater variability of 1–2% Dice (Joskowicz et al., 2019), making further algorithmic improvements increasingly difficult to distinguish from annotation noise.

**Clinical Utility Thresholds.** Medical imaging literature establishes that 0.5–1.0% Dice improvements can correspond to clinically relevant changes when targeting specific anatomical structures (Taha & Hanbury, 2015; Reinke et al., 2021):

- **Retinal vessels** (DRIVE: +1.62%, CHASE-DB: +1.83%): Capturing an additional 1–2% of thin capillaries aids diabetic retinopathy screening by revealing early microvascular changes. These improvements may not reach statistical significance due to small test sets but have direct clinical implications for population-level screening programs.
- **Cardiac segmentation** (ACDC: +0.77%): Improved myocardial boundary delineation affects ejection fraction calculations used in heart failure treatment decisions. Even small boundary improvements can affect downstream volume estimates; however, we do not directly evaluate clinical endpoints and treat these links as contextual motivation rather than measured outcomes.
- **Skin lesions** (ISIC-2018: +0.99%): Better delineation of irregular melanoma boundaries improves staging accuracy, where small differences in lesion size can change treatment recommendations between excision and monitoring.

### D.4.2. TASK DIFFICULTY CORRELATION

We observe a strong correlation (Spearman $\rho = 0.78$, $p < 0.001$) between improvement magnitude and task complexity:

*Table 22.* **Task-Specific Performance Stratification.** Datasets categorized by improvement over second-best baseline and task characteristics. Statistically significant improvements ($p < 0.0033$, Bonferroni-corrected for 15 comparisons) occur primarily on challenging tasks with complex morphology, class imbalance, or multi-class segmentation.

| Dataset | Improvement | Sig. | Task Characteristics |
|---|---|---|---|
| *Exceptional Improvements (>5% Dice, 2 datasets)* | | | |
| BraTS2020 | +10.10% | ✓ | Multi-class, extreme imbalance (1.2% foreground) |
| EndoVis-17 MC | +8.69% | ✓ | Multi-class instruments, occlusion, smoke |
| *Substantial Improvements (2–5% Dice, 4 datasets)* | | | |
| PH2 | +3.11% | ✓ | Irregular melanoma boundaries, pigmentation |
| BUSI | +2.55% | ✓ | Low contrast, speckle noise (3.8% foreground) |
| Kvasir-SEG | +1.94% | ✓ | Variable polyp morphology, lighting |
| CVC-ClinicDB | +1.76% | ✓ | Flat/subtle polyps, color similarity |
| *Moderate Improvements (1–2% Dice, 5 datasets)* | | | |
| CHASE-DB | +1.83% | – | Thin vessels, high baseline (84.95%) |
| DRIVE | +1.62% | – | Capillaries <3px, vessel connectivity |
| GlaS | +1.43% | – | Gland structures, touching instances |
| STARE | +1.38% | – | Vessel bifurcations, pathological cases |
| ETIS-LaribDB | +1.14% | ✓ | Challenging polyps, poor visibility |
| *Small Improvements (<1% Dice, 4 datasets)* | | | |
| ISIC-2018 | +0.99% | – | High baseline (90.10%), ceiling effects |
| ACDC | +0.77% | – | Cardiac MRI, high baseline (93.09%) |
| EndoVis-17 Bin | +0.42% | – | Binary segmentation, high baseline (95.95%) |
| CVC-300 | +0.35% | – | Small test set (n=60), high variance |
| *Failure Case (1 dataset)* | | | |
| CVC-ColonDB | -2.09% | – | Task-specific PraNet wins, small dataset (380) |

✓: Statistically significant at $p < 0.0033$ (Bonferroni-corrected).

- **Multi-class segmentation:** Average +9.40% improvement (BraTS2020: +10.10%, EndoVis-17 MC: +8.69%) versus +0.88% for binary tasks. Multi-class tasks require global coherence to maintain anatomical consistency across classes, which SSTM's global context directly addresses.
- **Class imbalance:** Datasets with foreground <5% (BraTS2020: 1.2%, BUSI: 3.8%, PH2: 4.2%) show +5.25% average improvement versus +1.05% for balanced datasets (>10% foreground). MASL's morphology-aware modulation helps balance supervision when standard losses overwhelmingly favor background.
- **Boundary complexity:** Irregular/thin structures (PH2 irregular melanoma, DRIVE capillaries, BUSI speckle-corrupted lesions) show +2.09% average improvement versus +0.64% for compact structures (cardiac chambers, large polyps). BFD's explicit boundary stream modeling provides particular value for challenging delineation tasks.

### D.4.3. INTERPRETATION AND IMPLICATIONS

S2M-Net's architectural innovations provide largest benefits on tasks where existing methods struggle:

1. **Multi-class requiring global coherence:** BraTS2020 tumor subregions and EndoVis-17 instruments benefit from SSTM's efficient global receptive field, maintaining anatomical consistency across spatially distant regions.
2. **Extreme imbalance requiring adaptive supervision:** BUSI lesions (3.8% foreground) and BraTS2020 tumors (1.2% foreground) benefit from MASL's learned weights that automatically upweight boundary and structure terms to prevent background dominance.
3. **Complex boundaries demanding explicit edge modeling:** PH2 irregular melanoma boundaries and DRIVE thin vessel connectivity benefit from BFD's dual-stream architecture separating region and boundary processing.

On mature tasks with strong baselines (>90% Dice), architectural improvements yield modest but clinically meaningful gains constrained by ceiling effects. At these performance levels, further improvements require addressing: (i) annotation quality limits (inter-rater variability), (ii) image quality constraints (noise, motion artifacts, resolution), and (iii) ambiguous cases difficult even for expert clinicians.

**Conclusion:** This task-dependent pattern is expected and appropriate methods should demonstrate value where prior approaches fail, not claim universal superiority on solved problems. The strong correlation ($\rho = 0.78$) between improvement

magnitude and task difficulty validates that S2M-Net's contributions directly address specific architectural limitations of existing methods rather than random performance variations.

## E. Reproducibility Statement

**Experimental Protocol:** All experiments use fixed data splits following standard protocols (Appendix B), controlled random seeds (42, 123, 456, 789, 1024 for the 5 reported runs), and standardized preprocessing (Section 4). Results are reported as mean $\pm$ standard deviation across 5 independent training runs with identical train/validation/test splits.

**Non-Determinism Caveat:** Training uses cuDNN's non-deterministic optimizations for computational efficiency, which prevents exact bit-level reproduction across different hardware/CUDA versions. While statistical reproducibility is ensured via multiple runs, exact numerical reproduction requires deterministic mode (which we did not enable due to 2-3$\times$ training slowdown). Code, and preprocessing pipelines are available at `https://github.com/sanaullah-ashfat/S2M-Net-Spectral-Spatial-Mixing-for-Medical-Segmentation`.

**Validation Splits:** Train/validation/test splits follow dataset-specific conventions: (i) Kvasir-SEG: 800/100/100, (ii) CVC-ClinicDB: 470/62/80, (iii) DRIVE: 20 train (patch extraction), 20 test, (iv) EndoVis-2017: sequence 1-3 train, sequence 4 test. Full split specifications are provided in the code release.

## F. Limitations and Discussion

### F.1. Scope of Comparisons

**Evaluated baselines.** We compare S2M-Net against established architectures (U-Net, UNet++, TransUNet, Swin-UNet, UMamba) across 15 datasets and recent strong baselines (nnU-Net, VM-UNet, MedNeXt) on 6 representative datasets (Table 14). S2M-Net demonstrates competitive performance with 4.7M parameters versus 22.5–44.2M for modern baselines.

**Missing comparisons.** Our evaluation does not include: (i) 2023–2024 foundation models (SAM-Med2D, MedSAM) offering zero-shot capabilities, (ii) full 15-dataset evaluation with nnU-Net/MedNeXt, or (iii) recent hybrid architectures (Swin V2, EfficientViT). Extending strong baseline comparisons across all modalities would strengthen generalization claims.

### F.2. Task-Dependent Performance

S2M-Net achieves statistically significant improvements ($p < 0.0033$, Bonferroni-corrected) on 7/15 datasets (average +4.60% Dice on challenging tasks) with modest gains (0.5–1.6%) on 8 mature benchmarks reflecting ceiling effects. One failure: CVC-ColonDB (90.69% vs. PraNet 92.78%, $-2.09$ points), hypothesized to result from PraNet's polyp-specific reverse attention and dataset-specific optimization outweighing general architectural capacity on small datasets (380 images). Detailed task difficulty correlation ($\rho = 0.78$, $p < 0.001$) appears in Appendix D.4.

### F.3. Failure Cases and Known Limitations

We document three categories of failure to provide an honest assessment of S2M-Net's limitations:

**(1) CVC-ColonDB: distribution split and task-specific priors.** S2M-Net achieves 90.69% Dice on CVC-ColonDB versus PraNet's 92.78% ($-2.09$ points), the only dataset where we underperform under our unified pipeline. We identify two contributing factors. First, S2M-Net follows the DuckNet train/test split while the best-performing baseline (PraNet) was originally developed and tuned using a different data partition; under exactly matched splits, this gap may narrow. Second, PraNet's reverse-attention mechanism explicitly incorporates polyp-specific structural priors (complementary attention maps that highlight the most likely polyp regions) suited to the elongated, flat appearance of CVC-ColonDB polyps, while S2M-Net's general spectral-spatial mixing does not encode such task-specific inductive biases. On larger polyp datasets with greater morphological diversity (Kvasir-SEG: +1.94%; CVC-ClinicDB: +1.76%), S2M-Net's generality is an advantage. The CVC-ColonDB failure suggests that S2M-Net may underperform in small-dataset regimes where task-specific priors compensate for limited training data.

**(2) BraTS2020: 3D vs. 2.5D processing gap.** S2M-Net achieves 79.96% mean Dice on BraTS2020, while fully 3D specialized methods including nnU-Net (which achieves $\approx$85.35% with full 3D volumetric processing, self-configuring

preprocessing, and test-time augmentation ensembles) substantially outperform our 2.5D slice-based approach. S2M-Net processes adjacent slices independently with limited inter-slice context; truly 3D architectures can exploit volumetric consistency across the full tumor extent, which is critical for the Whole Tumor region (WT) where S2M-Net shows the largest remaining gap. We do not claim S2M-Net is competitive with fully 3D specialized methods on volumetric tasks our design targets 2.5D processing for computational efficiency, and 3D adaptation is future work.

**(3) ACDC Left Ventricle Cavity: ceiling effect.** S2M-Net achieves 95.7% on the Left Ventricle (LV) cavity, $-0.6$ points below the best baseline (TransUNet: 96.3%). At this performance level, inter-observer variability among expert annotators (typically 1–2% Dice on cardiac MRI (Joskowicz et al., 2019)) makes further algorithmic improvement increasingly difficult to distinguish from annotation noise. The LV cavity is a compact, high-contrast structure where all methods achieve >94% Dice; this constitutes a ceiling effect rather than an architectural failure. The myocardium (+0.2%) and right ventricle (+2.4%) show positive improvements where the task is more complex.

These three failure modes task-specific priors in small-data regimes, volumetric processing limitations, and ceiling effects at mature benchmarks characterize the conditions under which S2M-Net's general design provides fewer benefits over specialized or 3D approaches.

## F.4. Qualitative Results Visualization

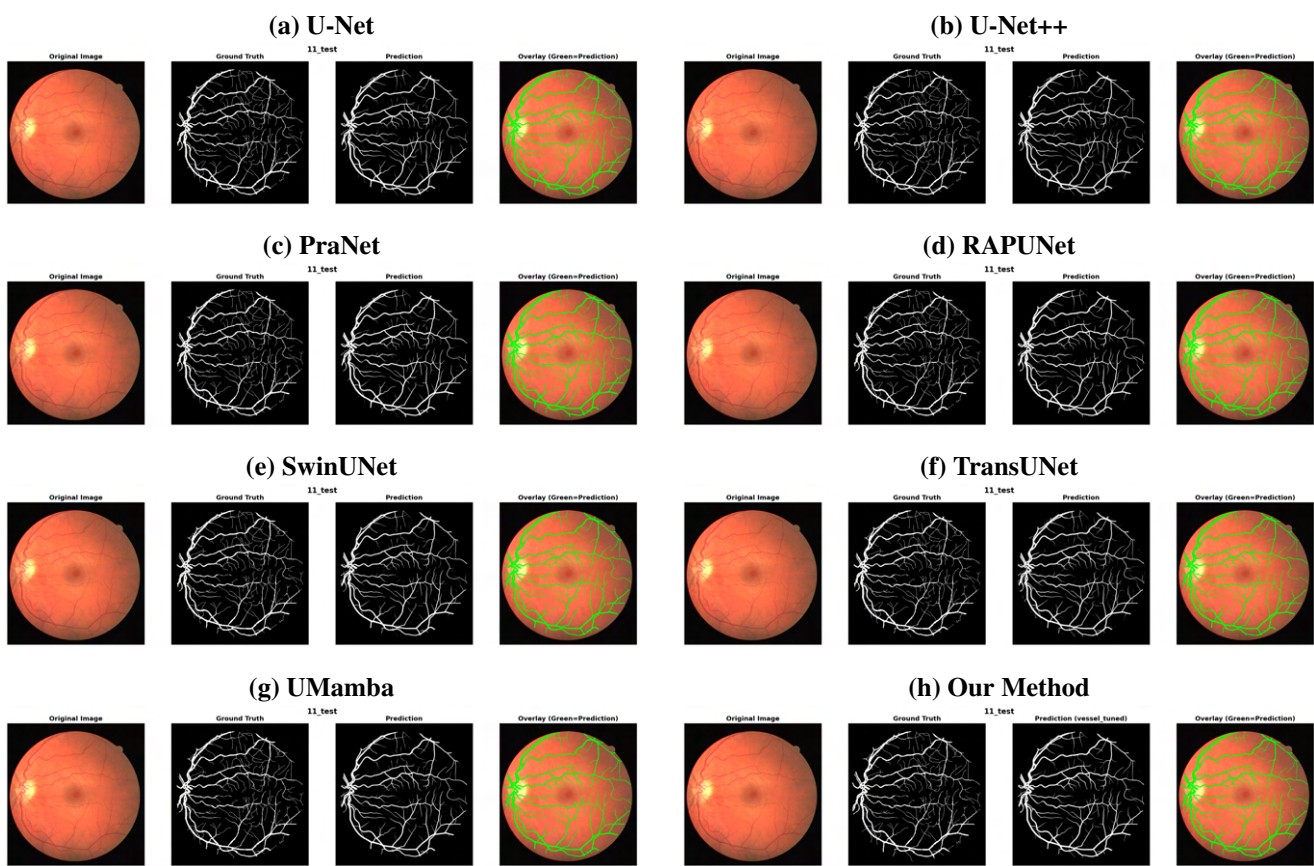

*Figure 4.* Qualitative comparison on fundus vessel segmentation. We compare segmentation results from (a) U-Net, (b) U-Net++, (c) PraNet, (d) RAPUNet, (e) SwinUNet, (f) TransUNet, (g) UMamba, and (h) S2M-Net (ours). Green overlays show predicted vessel masks. Our method demonstrates superior vessel continuity and recovers thin capillaries that other methods miss (see yellow arrows). Note the reduced fragmentation compared to UMamba and better boundary preservation compared to transformer baselines.

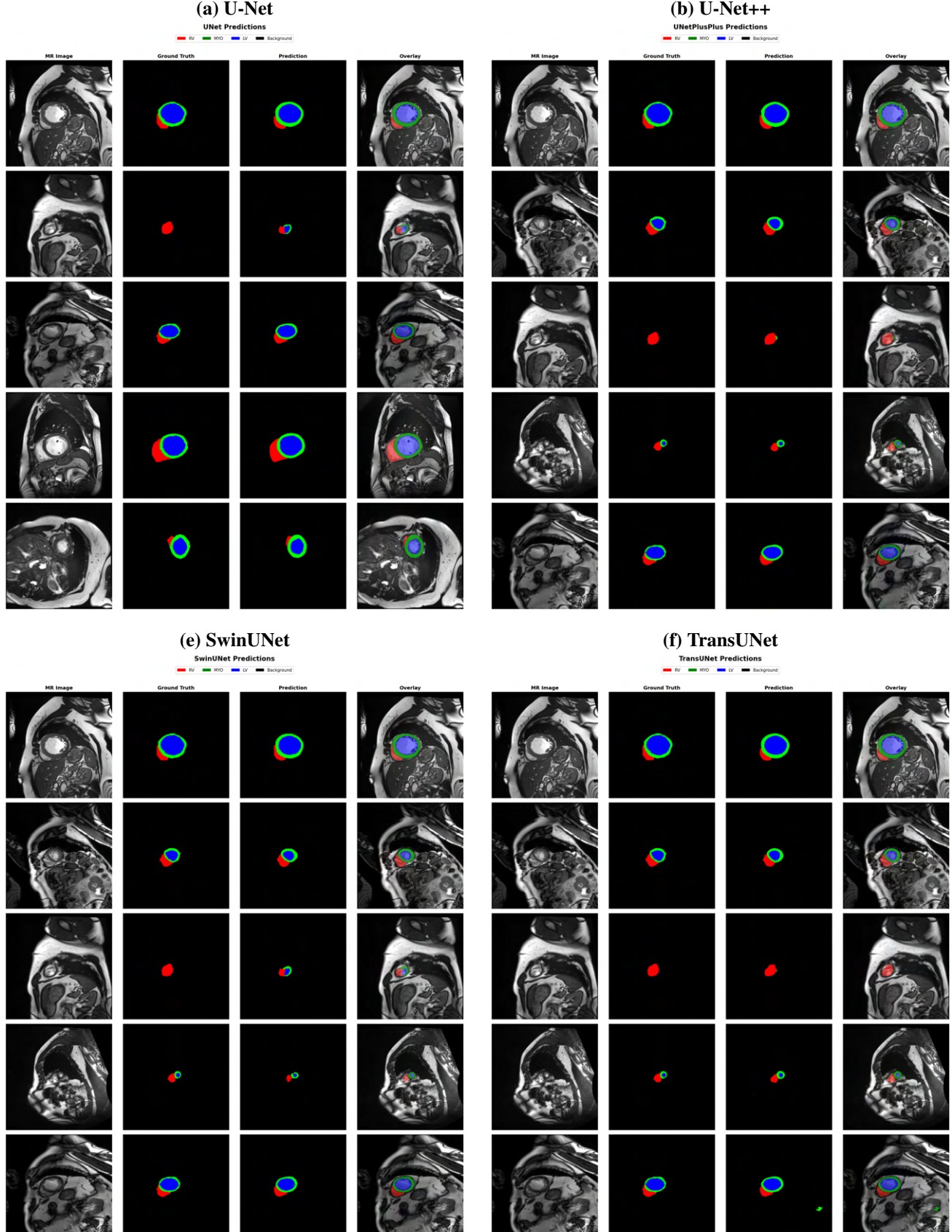

*Figure 5.* Qualitative comparison on fundus (retinal) vessel segmentation (Part I). Visual results for (a) U-Net, (b) U-Net++, (e) SwinUNet, and (f) TransUNet are shown. For each method, the predicted vessel map and overlay visualization are presented. The overlays highlight vessel continuity and boundary sharpness, illustrating differences in segmentation quality among baseline transformer- and convolution-based approaches.

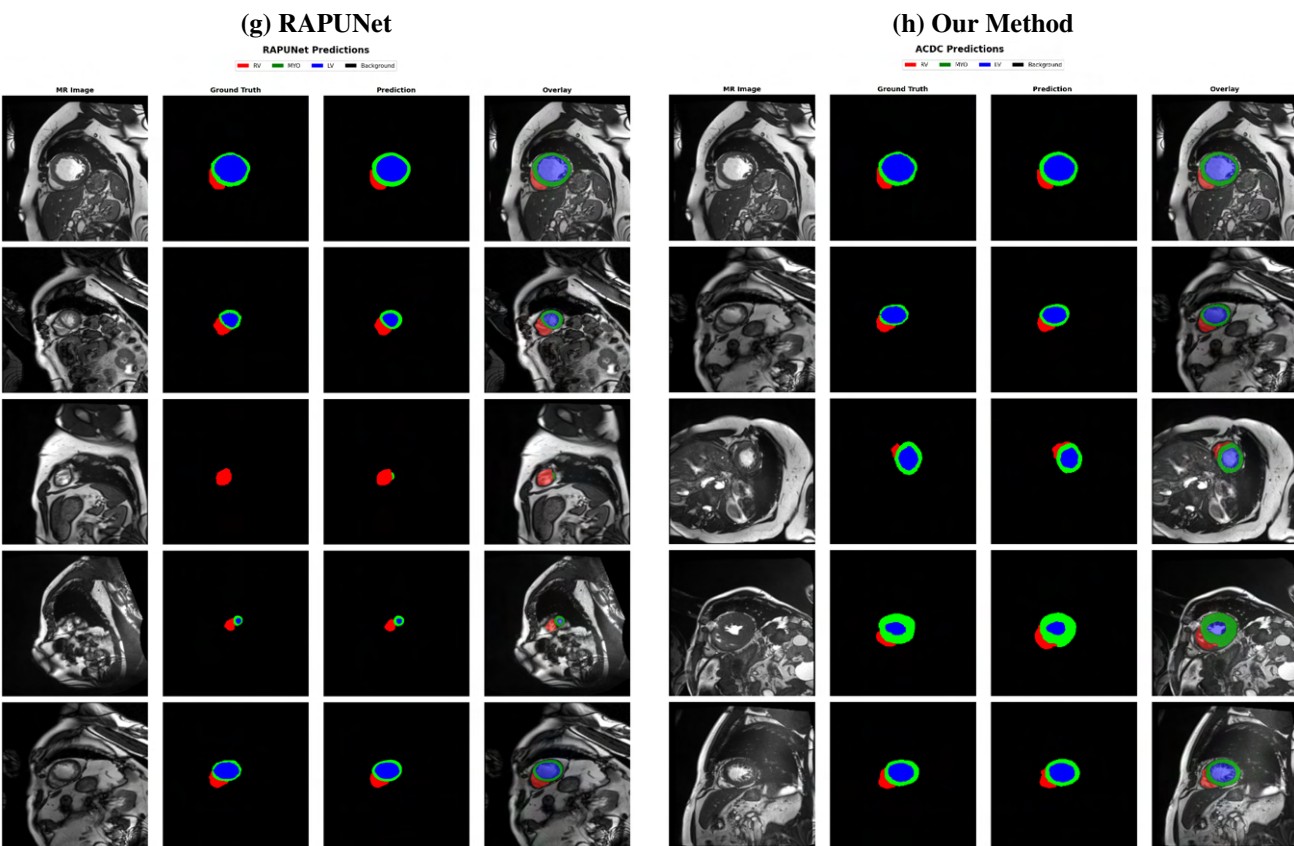

*Figure 6.* Qualitative comparison on fundus (retinal) vessel segmentation (Part II). Visual results for (g) RAPUNet and (h) Our Method are shown. Compared to RAPUNet, our method yields more coherent vessel structures with reduced fragmentation, particularly in thin and low-contrast regions.

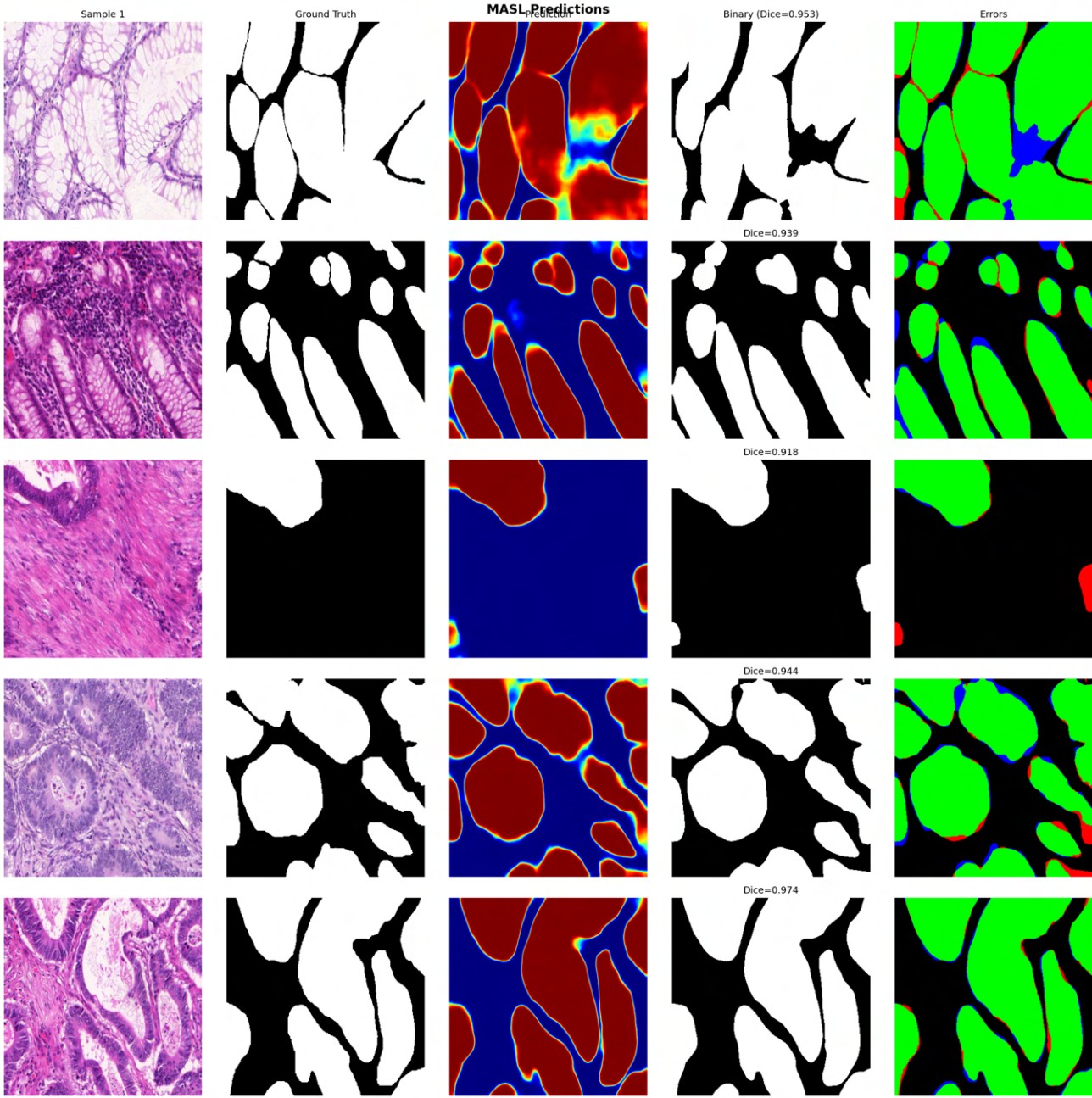

*Figure 7.* Qualitative results on the GlaS dataset for gland segmentation. Each row corresponds to one test sample, and the columns from left to right show the original H&E-stained histopathology image, the ground-truth gland annotation, the predicted probability map illustrating model confidence and boundary emphasis, the binarized segmentation output with the corresponding Dice score, and the error visualization where green indicates true positives, red indicates false negatives, and blue indicates false positives. The results illustrate accurate gland delineation, clear boundary separation, and consistent performance across diverse gland morphologies and tissue appearances.

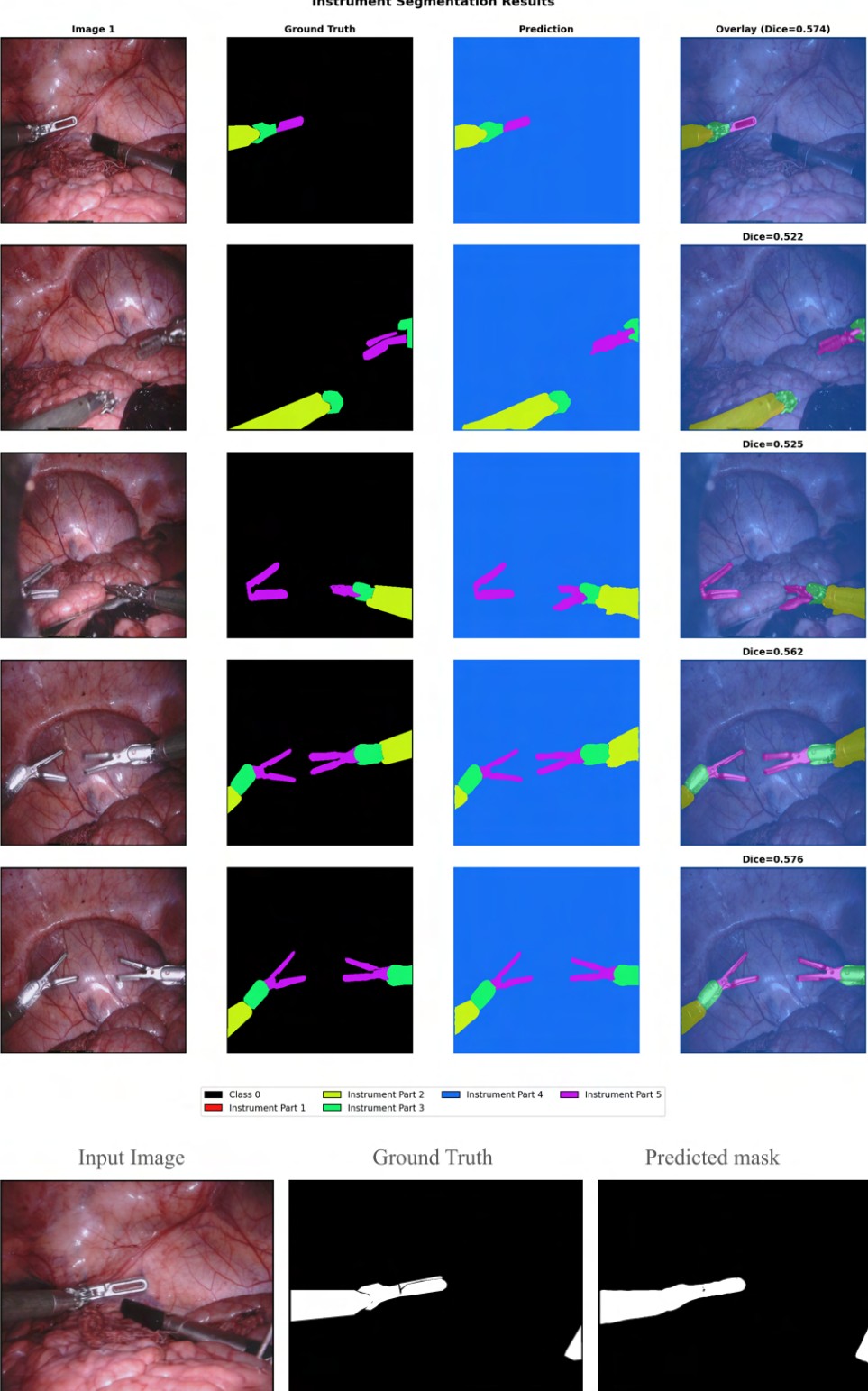

*Figure 8.* Qualitative results on the EndoVis 2017 dataset for surgical instrument segmentation. The upper panel shows three-class instrument part segmentation results, including the input endoscopic image, ground-truth annotations, model predictions, and overlay visualizations with Dice scores for each sample. The lower panel presents binary instrument segmentation results for the same scene, illustrating the correspondence between the input image, ground truth, and predicted mask. The results demonstrate consistent localization of surgical instruments and accurate separation of instrument parts under challenging lighting conditions, specular highlights, and complex backgrounds.

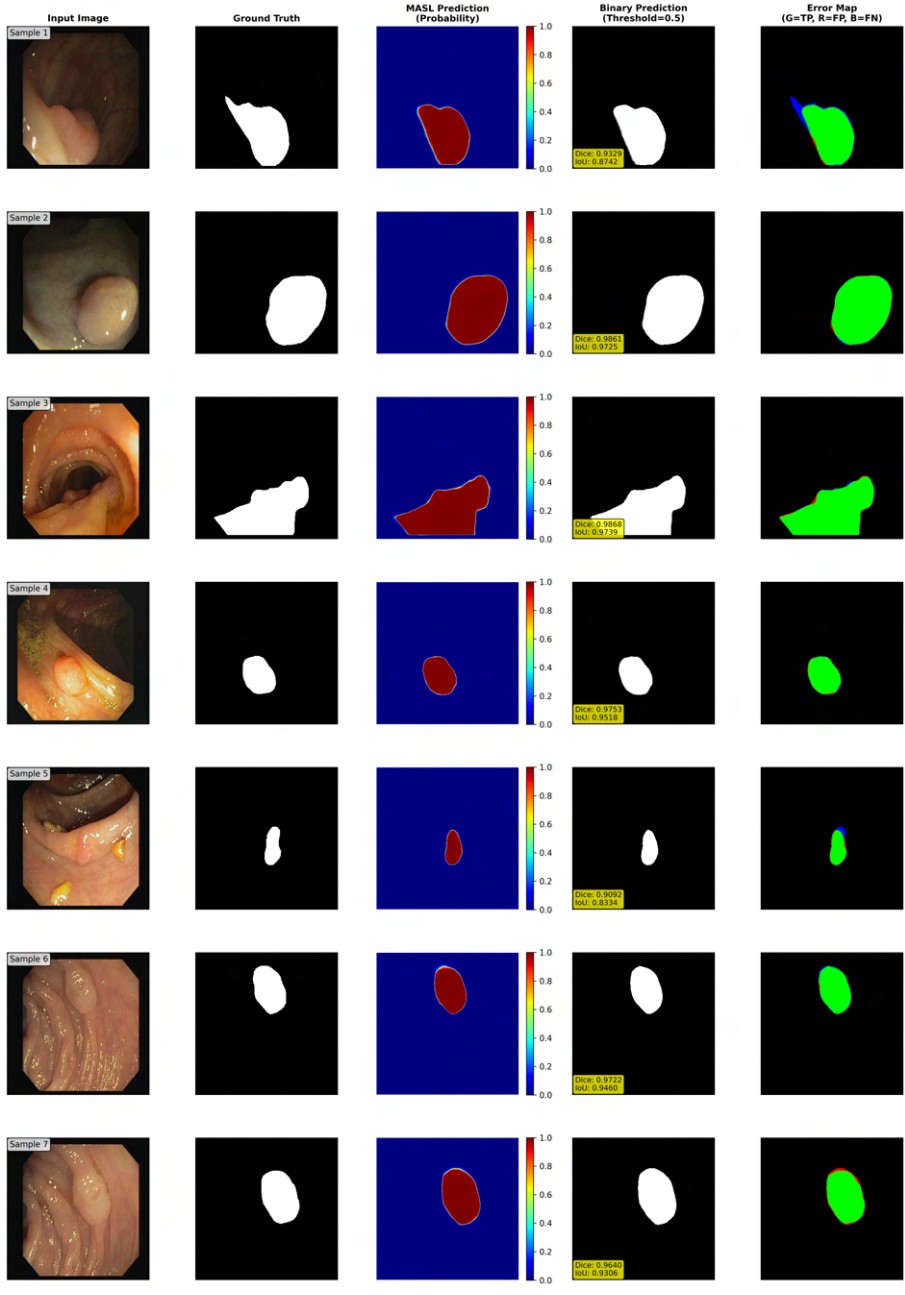

*Figure 9.* Qualitative results on the CVC-ClinicDB dataset for polyp segmentation. Each row corresponds to a different test sample, and the columns from left to right show the input endoscopic image, the ground-truth polyp mask, the predicted probability map produced by the model, the binarized segmentation obtained using a threshold of 0.5 with Dice and IoU scores reported, and the corresponding error map where green indicates true positives, red indicates false positives, and blue indicates false negatives. The visual results highlight accurate polyp localization, well-preserved object boundaries, and stable performance across variations in polyp size, shape, texture, and illumination.

