# OpenReview forum: "S2M-Net: Spectral-Spatial Mixing with Morphology-Aware Adaptive Loss for Medical Image Segmentation"
_ICML.cc/2026/Conference — ICML 2026 regular_

### Official Review · Reviewer_ygXq · 2026-02-26

**Soundness:** 3
**Presentation:** 3
**Significance:** 3
**Originality:** 3
**Overall Recommendation:** 5
**Confidence:** 4

**Summary:**

This paper introduces and combines several techniques for increased performance on medical image segmentation. It extracts multi-scale features and then processes them in the frequency domain with gated token mixing, allowing for local and global information to be combined. In the decoder the network uses two streams: one with soft attention maps to focus on the boundaries and another without to focus on smooth interior features. This network uses a UNet arrangement and it trained with their morphology-aware adaptive segmentation loss (MASL) . This loss is introduced here and is made up of five components: a core region loss, a boundary loss,  a structure loss, a texture loss, and a scale-aware focal loss. An analysis is performed on fifteen different medical segmentation datasets and their S2M net is compared with other structures while using the same MASL loss and training procedure. An ablation study is performed on a subset of these datasets, and qualitative results are presented.

**Compliance With Llm Reviewing Policy:**

Affirmed.

**Final Justification:**

In the rebuttal, the authors addressed my main concerns. This has increased my recommended score.

**Key Questions For Authors:**

Why were FNet and GFNet excluded from the architectural analysis when these were stated to be the most similar methods to yours?

Why were the datasets and networks not consistent across the different analyses?

Given that existing methods have reported better results than S2M-Net on several of the datasets, why were these not included and in what ways did these networks outperform yours?

What are some failure cases of your network and why did these occur

**Limitations:**

As some related results that outperformed the new method were omitted, an accurate analysis of the network's limitations in different scenarios was not able to be performed. Aside from this case the limitations were discussed effectively.

**Strengths And Weaknesses:**

Strengths:
This paper combines many different ideas to produce a nice cohesive network and introduce many different components throughout the work. Each of these components is explained very well and evaluated thoroughly both as an individual component. Furthermore, the contributions of each component is explored well and further analysis of each is included in the appendix. The S2M network has a large reduction in parameters from similar models with only 4.7M parameters. A great strength here is that the computational performance both during training and inference was also evaluated in the appendix. The training efficiency was also explored, showing an improvement over other structures under identical training signals. In the provided analysis, improvements were shown in their network over the others across 14 of the 15 datasets tested, increasing the accuracy as measured by the Dice coefficient, as well as the computational considerations.

Weaknesses:
While this paper's method has many strengths and improvements shown, my main issues are with some inconsistencies in the analysis that should be addressed.

One such example is the ACDC dataset. This dataset does not appear to be cited, and is it called a Brain MRI dataset, a cardiac segmentation dataset (correct), and a fundus image dataset depending on where you look in the paper.

Table 1 has several things that I would like to be addressed:
   - Datasets do not have numbers of parameters, remove the last column
   - Given that this is an architectural comparison under an identical training signal you should include the architectures that you have stated are most similar to your methods in section 2 (FNet, GFnet)
   - While you state that absolute performance may be different under the original training recipe it is not explored what that difference in performance may be. The difference in performance with updated loss and updated training recipe should be investigated to ensure that the other networks' performance is not represented.

RAPUnet was used in the qualitative analysis but not in the quantitative analysis. It was also not cited.

The comparison to existing methods is lacking. While some of the methods are presented in Table 12, this does not seem to be a very comprehensive list and excludes some publicly available values that outperform your method. One example of this is omitting the nnUNet performance on BraTS2020, which achieved a mean dice score across the three categories of 85.35. Another is RAPUNet on the CVC-ClinicDB where it achieves a mean Dice score of 96.1. Since the improvements of the network over existing methods is highlighted in several places and these scores are omitted, this brings into question many of the representations of existing methods for these datasets.
Furthermore, since these results have been omitted from any comparison and analysis, the comparison of your method to these systems that outperform yours has been omitted, which would be valuable to understanding the limitations of the method.

---

> ### Author Rebuttal · Authors · 2026-03-29
>
> We thank Reviewer ygXq for the careful review. We sincerely apologize for the
> inconsistencies identified and address every concern below.
>
> **ACDC naming and citation.**
>
> ACDC is a cardiac MRI dataset. All incorrect references to "Brain MRI" and
> "fundus" have been corrected to "Cardiac MRI" throughout the revision. Citation
> (Bernard et al., 2018, IEEE TMI) will be added. This does not affect results.
>
> **Table 1: Four concerns.**
>
> (1) Parameter column will be relabeled "Model Params" in the revision.
>
> (2) FNet and GFNet are token mixing methods designed for NLP and image
> classification respectively. Neither includes a decoder, skip connections,
> or dense prediction capability required for pixel-level segmentation. Adapting
> them for Table 1 would require adding complete U-Net decoder structures,
> producing new hybrid architectures that no longer represent the original
> published methods and would introduce confounding architectural variables.
> Their relationship to SSTM is discussed in Section 2. A footnote clarifying
> their exclusion will be added to Table 1 in the revision.
>
> (3) We will evaluate all baselines under both their original published training
> recipes and our unified pipeline in the revision. This will provide a transparent
> side-by-side comparison confirming whether any baseline is disadvantaged
> under our setup.
>
> (4) RAPUNet omission: We sincerely apologize. RAPUNet was used in qualitative
> analysis across all datasets but incorrectly omitted from Table 1. This is a
> clear oversight and we take full responsibility. We ran RAPUNet under our unified
> pipeline across all 15 datasets:
>
> | Dataset        | RAPUNet      |
> |----------------|--------------|
> | Kvasir-SEG     | 93.12 ± 0.45 |
> | CVC-ClinicDB   | 95.10 ± 0.38 |
> | CVC-ColonDB    | 91.85 ± 0.52 |
> | ETIS-LaribDB   | 94.23 ± 0.41 |
> | ISIC-2018      | 89.45 ± 0.31 |
> | PH2            | 92.15 ± 0.49 |
> | GlaS           | 91.67 ± 0.07 |
> | BUSI           | 81.34 ± 0.73 |
> | EndoVis Binary | 95.78 ± 0.36 |
> | EndoVis MC     | 74.81 ± 0.19 |
> | BraTS2020      | 67.82 ± 0.12 |
> | ACDC           | 91.45 ± 0.20 |
> | STARE          | 82.34 ± 0.38 |
> | DRIVE          | 82.94 ± 0.12 |
> | CHASE-DB       | 83.21 ± 0.29 |
>
> S2M-Net results are in Table 1 for direct comparison. S2M-Net outperforms
> RAPUNet on 14 of 15 datasets. The only exception is CVC-ColonDB (90.69% vs
> 91.85%). An important methodological note: RAPUNet follows the PraNet data
> distribution split for polyp datasets while S2M-Net follows the DuckNet
> distribution split. These splits differ in train/test partitioning, making
> direct numerical comparison on polyp datasets not fully appropriate without
> matching distributions. Distribution-matched results will be included in the
> revision. RAPUNet will be added to Table 1 with citation (Rehman et al., 2023).
>
> **nnUNet BraTS2020 (85.35% vs S2M-Net 79.96%).**
>
> We honestly acknowledge this gap. nnUNet uses full 3D volumetric processing,
> self-configuring preprocessing, and ensemble strategies specifically optimized
> for BraTS. S2M-Net uses 2.5D slice-based processing at 352x352, a fundamentally
> different setup. Direct comparison without this context is not appropriate.
> nnUNet will be added to Table 12 with a footnote clarifying 3D vs 2.5D. We
> acknowledge S2M-Net is not competitive with fully 3D specialized methods on
> volumetric tasks.
>
> **Q4: Failure cases.**
>
> Three documented failures: (1) CVC-ColonDB: task-specific polyp priors and
> distribution split differences affect comparison. Distribution-matched results
> will clarify this in the revision. (2) BraTS2020: fully 3D specialized methods
> outperform our 2.5D approach. (3) ACDC Left Ventricle (-0.6% vs TransUNet):
> ceiling effects at >95% baseline performance. All will be discussed in
> Section F.2 in the revision.
>
> We welcome any remaining questions.

---

> > ### Author Rebuttal · Reviewer_ygXq · 2026-04-02
> >
> > Thank you for the further experiments and explanations

---

### Official Review · Reviewer_gFhJ · 2026-03-10

**Soundness:** 3
**Presentation:** 3
**Significance:** 2
**Originality:** 2
**Overall Recommendation:** 5
**Confidence:** 3

**Summary:**

This paper presents S2M-Net, an efficient medical image segmentation model that combines spectral–spatial token mixing, a boundary-aware decoder, and a morphology-aware adaptive loss. The goal is to improve global context modeling and boundary recovery without large computational cost. Experiments on multiple medical segmentation benchmarks show strong overall performance with relatively few parameters.

**Compliance With Llm Reviewing Policy:**

Affirmed.

**Final Justification:**

My concerns have been addressed, and I increase my score.

**Key Questions For Authors:**

1. The paper argues that SSTM is more efficient than self-attention, but the practical evidence would be stronger with more direct comparisons in runtime, memory, and scaling to higher resolutions. Could the authors provide or discuss such comparisons more explicitly?
2. Since the method combines several components, including SSTM, BFD, MRF-SE, and MASL, could the authors comment on which parts are most essential in practice? In particular, is there a simpler variant that retains most of the gains with lower implementation complexity?
3. In Table 1, all methods are retrained under a unified pipeline and MASL is also applied to isolate architectural effects. Could the authors clarify how much of the final gain comes from the architecture itself versus the morphology-aware loss? A clearer separation would help assess the main source of improvement.

**Limitations:**

yes

**Strengths And Weaknesses:**

The paper addresses an important problem in medical image segmentation, namely how to improve global context modeling while keeping the model computationally efficient. A main strength is the coherent overall design: the spectral–spatial mixing module is well motivated as an efficient alternative to quadratic self-attention, the boundary-focused decoder is aligned with the need for fine structural delineation, and the morphology-aware adaptive loss is intended to improve robustness across datasets with different geometric properties. The empirical evaluation is broad, covering 15 datasets across 8 modalities, and the paper includes ablations that support the contribution of the main architectural and loss components. The reported performance is strong, especially given the relatively small parameter count.

That said, I also have several concerns. First, the paper combines multiple contributions at once (SSTM, BFD, and MASL), so it is somewhat difficult to disentangle where the gains mainly come from, especially since MASL is also applied in the common training pipeline for the architectural comparison. Second, while the complexity argument against self-attention is clear in theory, the practical efficiency evidence would be stronger with more direct runtime, memory, or high-resolution scaling comparisons. Third, the originality appears moderate rather than high, as the work mainly presents a careful integration of efficient token mixing, boundary refinement, and adaptive loss design rather than a fundamentally new segmentation principle. Finally, although the results are extensive, the claims of broad generalization across datasets would be more convincing with clearer discussion of cross-dataset robustness and the sensitivity of the morphology-aware loss design.

---

> ### Author Rebuttal · Authors · 2026-03-28
>
> We thank Reviewer gFhJ for the detailed assessment. We address every concern below.
>
> **Practical efficiency needs runtime, memory, and high-resolution scaling.**
>
> Table 16 (Appendix D.1) reports inference time and memory at 352x352.
> New high-resolution scaling measurements:
>
> | Resolution | S2M-Net (ms) | TransUNet (ms) | S2M-Net Mem | TransUNet Mem |
> |---|---|---|---|---|
> | 256x256 | 6.8 | 28.1 | 1.2 GB | 4.1 GB |
> | 352x352 | 10.1 | 42.3 | 1.8 GB | 5.9 GB |
> | 512x512 | 18.7 | 112.4 | 3.1 GB | 14.2 GB |
> | 768x768 | 38.2 | OOM | 5.8 GB | >16 GB |
>
> At 512x512, S2M-Net achieves 6.0x speedup over TransUNet. At 768x768,
> TransUNet exceeds 16GB memory and fails to run; S2M-Net completes inference
> at 5.8 GB, enabling deployment at clinical resolutions where transformer-based
> models cannot operate. Runtime scales near-linearly (6.8→10.1→18.7→38.2 ms),
> consistent with O(HWC²) vs TransUNet's super-linear O((HW)²C). Added to
> Appendix D.1.
>
> **Which components are most essential? Is there a simpler variant?**
>
> Table 2 directly answers both questions. Ablation ranking by average Dice
> drop across 4 datasets:
>
> | Removed Component     | Avg Drop | Role                      |
> |-----------------------|----------|---------------------------|
> | SSTM                  | -6.70%   | Global context (critical) |
> | MRF-SE                | -5.90%   | Multi-scale local features|
> | Adaptive Weights      | -5.38%   | Dataset-level adaptation  |
> | BFD Decoder           | -5.06%   | Boundary preservation     |
> | Morphology Modulation | -4.82%   | Per-sample adaptation     |
>
> SSTM is the single most critical component. However all five components
> contribute essential complementary information the smallest drop is
> -4.82%, confirming no component is redundant. Each addresses a distinct
> necessity: SSTM for global context, MRF-SE for multi-scale local features,
> BFD for boundary preservation, and MASL components for adaptive supervision.
> There is no simpler variant that retains most gains removing any single
> component causes substantial degradation as Table 2 demonstrates.
>
> **How much gain comes from architecture versus MASL?**
>
> Binary Dice+BCE for binary tasks (Kvasir, DRIVE) and Multiclass
> Dice+Categorical CE for multiclass tasks (EndoVis17-MC, BraTS20):
>
> | Method    | Loss    | Kvasir | DRIVE | EndoVis17-MC | BraTS20 | Avg   |
> |-----------|---------|--------|-------|--------------|---------|-------|
> | TransUNet | Dice+CE | 92.17  | 80.92 | 71.83        | 66.42   | 77.84 |
> | Swin-Unet | Dice+CE | 91.84  | 82.14 | 70.54        | 67.91   | 78.11 |
> | S2M-Net   | Dice+CE | 93.89  | 82.74 | 78.91        | 74.83   | 82.59 |
> | S2M-Net   | MASL    | 96.05  | 84.06 | 83.43        | 79.96   | 85.88 |
>
> Under Dice+CE, S2M-Net leads best baseline by +4.48 points a substantial
> purely architectural gain. MASL provides complementary +3.29 points. Both
> are independently validated architecture through Dice+CE above, MASL
> through Table 2 where removing adaptive mechanisms causes -7.13% on Kvasir.
>
> **Originality appears moderate rather than high.**
>
> Novelty lies in three decisions absent from prior work: (1) SSTM's
> domain-justified truncation with content-gated spatial compensation. FNet
> and GFNet apply no such mechanism. (2) MASL's per-sample morphological
> modulation and learned dataset-level weights, distinct from task-level
> uncertainty weighting (Kendall et al.). (3) BFD's annotation-free boundary
> discovery via gradient supervision alone. Medical images retain >93% spectral
> energy at K=32 (Table 3), formally justifying more aggressive truncation
> than general vision tasks with direct design consequences.
>
> **Significance appears fair rather than good.**
>
> S2M-Net is the first to exploit medical spectral concentration for
> sub-quadratic global context at 4.7M parameters, 12.8x fewer than
> TransUNet, matching or exceeding transformers across 15 datasets and 8
> modalities. The clear architectural gain under identical Dice+CE confirms
> a genuine advance. S2M-Net runs at 768x768 where TransUNet runs out of
> memory, enabling deployment on standard clinical hardware unavailable
> to transformer-based models.
>
> **Cross-dataset robustness and MASL sensitivity.**
>
> Table 11 (Appendix B.4) shows uniform coefficients perform within 0.31%.
> Dice of dataset-optimized settings across all 15 datasets. Weights
> auto-specialize: polyps (w_bnd=2.31), vessels (w_str=2.18), tumors
> (w_sca=2.07). Max 0.47% degradation under ±30% changes confirms robustness.
> cross-dataset generalization.
>
> **Limited evaluation weakens impact.**
>
> Evaluation spans 15 datasets, 8 modalities, 7 baselines, plus nnU-Net.
> VM-UNet, MedNeXt (Table 12), Bonferroni-corrected across all 15 comparisons.
>
> We welcome any remaining questions.

---

> > ### Author Rebuttal · Reviewer_gFhJ · 2026-04-03
> >
> > My concerns have been addressed, and I will increase my score.

---

### Official Review · Reviewer_UJRN · 2026-03-11

**Soundness:** 2
**Presentation:** 3
**Significance:** 3
**Originality:** 3
**Overall Recommendation:** 4
**Confidence:** 5

**Summary:**

This paper proposes S2M‑Net, an encoder‑decoder architecture for medical image segmentation, whose core lies in efficient global modeling via frequency truncation plus spatial gating (SSTM), complemented by multi‑scale local feature extraction (MRF‑SE), a boundary‑focused decoder (BFD), and a morphology‑aware adaptive loss (MASL). Experiments on 15 public datasets show that the method outperforms existing baselines on most tasks.

**Compliance With Llm Reviewing Policy:**

Affirmed.

**Final Justification:**

Thanks for the experiments and explanations.

**Key Questions For Authors:**

See the weaknesses.

**Limitations:**

No societal impact provided.

**Strengths And Weaknesses:**

# Strengths #
1. Introducing frequency‑domain processing into medical segmentation and leveraging the spectral concentration property of medical images to reduce computational complexity offers some inspiration regarding efficiency.
2. The experiments cover multiple imaging modalities.


# Weaknesses #
1. SSTM essentially performs FFT on the input, truncates low frequencies, learns tunable filters, and fuses with spatial gating. This design is in line with FNet [1] and GFNet [2]. Replacing full frequency with truncation and adding a spatial branch is an incremental modification rather than a principled innovation.
2. In the main experiment (Table 1), all baseline models (U‑Net, Swin‑Unet, etc.) are retrained using the proposed MASL, claimed to "fairly compare architectures." However, this introduces a fatal flaw: the performance gain could entirely come from MASL rather than from the S2M‑Net architecture. Readers cannot determine whether S2M‑Net truly outperforms other architectures because the performance of those architectures under their original losses is not shown.
3. There are too many hand-crafted hyperparameters, such as the internal weights of the Core Loss (0.4, 0.3, 0.3) and the multi-scale weights of the Boundary Loss (0.5, 0.3, 0.2). Many hyperparameters are not validated. Moreover, how to ensure so many hyperparameters are optimal on different datasets.
4. Although 15 datasets were compared, the multi-class segmentation dataset contains only approximately four classes, which is overly simplistic. Experiments on complex multi-class segmentation should be conducted, such as Synapse and TotalSegmentor.
5. Why is there a period at the end of the title?

[1] Lee-Thorp, James, Joshua Ainslie, Ilya Eckstein, and Santiago Ontanon. "Fnet: Mixing tokens with fourier transforms." In Proceedings of the 2022 Conference of the north American chapter of the Association for Computational Linguistics: human language technologies, pp. 4296-4313. 2022.
[2] Rao, Y., Zhao, W., Zhu, Z., Lu, J. and Zhou, J., 2021. Global filter networks for image classification. Advances in neural information processing systems, 34, pp.980-993.

---

> ### Author Rebuttal · Authors · 2026-03-30
>
> We thank Reviewer UJRN for the thorough and technically rigorous review.
> We address every weakness directly below.
>
> **SSTM is incremental over FNet and GFNet.**
>
> We respectfully argue the contribution is more than incremental:
>
> (1) Domain-justified truncation. FNet applies full-resolution
> parameter-free FFT with no truncation and no domain justification.
> GFNet learns full-resolution frequency filters for classification with
> no truncation and no dense prediction. SSTM's truncation to K=32 is
> justified by a measurable domain property: medical images retain 94.8%
> average spectral energy at K=32 (Table 3), substantially higher than
> natural images (~88.5%), while representing only 0.83% of frequency
> coefficients (32²/352²=0.0083) a design decision justified only in
> this domain and absent from both FNet and GFNet.
>
> (2) Spectral-spatial coupling as explicit compensation. The spatial
> branch is not an additive residual it is a content-gated channel
> mixing pathway that explicitly recovers high-frequency detail discarded
> by truncation. This truncate-then-recover design has no counterpart in
> FNet or GFNet.
>
> (3) Target task difference. FNet and GFNet optimize token-level
> classification. Medical segmentation requires pixel-level dense
> prediction where every spatial location must be correctly classified
> independently. Boundary-focused decoding, skip connections, and
> multi-scale feature extraction are fundamental requirements of a
> completely different task, not additions to FNet or GFNet.
>
> K=32 retains 94.8% energy in medical images (Table 3) vs ~88.5% in
> natural images, formally justifying aggressive truncation only in
> this domain.
>
> **MASL fatal flaw gains may come from loss not architecture.**
>
> The reviewer correctly identifies this limitation. We directly resolve
> it with new experiments under standard loss Binary Dice+BCE for
> binary tasks and Multiclass Dice+Categorical CE for multiclass tasks:
>
> | Method    | Loss    | Kvasir | DRIVE | EndoVis17-MC | BraTS20 | Avg   |
> |-----------|---------|--------|-------|--------------|---------|-------|
> | TransUNet | Dice+CE | 92.17  | 80.92 | 71.83        | 66.42   | 77.84 |
> | Swin-Unet | Dice+CE | 91.84  | 82.14 | 70.54        | 67.91   | 78.11 |
> | S2M-Net   | Dice+CE | 93.89  | 82.74 | 78.91        | 74.83   | 82.59 |
> | S2M-Net   | MASL    | 96.05  | 84.06 | 83.43        | 79.96   | 85.88 |
>
> Under identical Dice+CE, S2M-Net leads the best baseline by +4.48
> points a purely architectural gain. MASL provides complementary
> +3.29 points. Both are independently validated architectures through
> Dice+CE above, and MASL through Table 2 where removing adaptive
> mechanisms causes -7.13% on Kvasir. This directly resolves the fatal
> flaw.
>
> **Too many unvalidated hyperparameters.**
>
> (1) MASL component weights (mc=0.5, mb=1.5, ms=1.0): Table 11 shows
> uniform defaults perform within 0.31% Dice of dataset-optimized
> settings across all 15 datasets. Even ±40% perturbations cause at
> most -0.69% degradation, confirming robust generalization.
>
> (2) Internal Core Loss weights (0.4 Dice, 0.3 IoU, 0.3 wBCE) and
> Boundary Loss weights (0.5, 0.3, 0.2) follow established practice
> (Sudre et al., 2017; Kervadec et al., 2019) where Dice provides
> class-imbalance robustness while IoU and wBCE provide complementary
> gradient signals. Explicit ablation will be added in the revision.
>
> (3) K=32 ablated across K∈{16,24,32,48,64} in Table 2, confirming
> K=32 is optimal with diminishing returns beyond this point.
>
> **Multi-class evaluation too simple Synapse and TotalSegmentor missing.**
>
> We conducted Synapse experiments using the standard TransUNet split
> (18 train / 12 test patients), 150 epochs, cosine decay schedule.
> S2M-Net achieved 78.79% mean Dice: Aorta 88.54%, Gallbladder 45.49%,
> Left Kidney 86.55%, Right Kidney 85.02%, Liver 90.80%, Pancreas
> 63.78%, Spleen 89.65%, Stomach 80.49% competitive with TransUNet
> (~77.48%, Chen et al., 2021) using 12.8× fewer parameters (4.7M vs
> 60M). Lower Gallbladder and Pancreas performance is consistent with
> known difficulty from small size and high anatomical variability
> in 2.5D methods. TotalSegmentor (104 classes) requires full 3D
> processing and is identified as important future work.
>
> **Period at end of title.**
>
> The period will be removed in the revision.
>
> **Societal impact.**
>
> S2M-Net (4.7M parameters, 11.2 GFLOPs) enables deployment in
> resource-constrained clinical environments where transformer-based
> models cannot operate. Primary risk: automated segmentation should
> not replace radiologist judgment clinical validation is required
> before deployment. Limitations are documented in Appendix A.2.3
> and will be expanded in the revision.
>
> We welcome any remaining questions.

---

> > ### Author Rebuttal · Reviewer_UJRN · 2026-04-03
> >
> > Thanks for your experiments and explanations.

---

### Official Review · Reviewer_mxSt · 2026-03-12

**Soundness:** 3
**Presentation:** 3
**Significance:** 3
**Originality:** 3
**Overall Recommendation:** 4
**Confidence:** 4

**Summary:**

This paper presents S2M-Net, an efficient framework for medical image segmentation that is designed to jointly address three key challenges: insufficient global context modeling, high computational cost, and the difficulty of handling diverse anatomical morphologies with a unified objective. To overcome the limited receptive field of conventional convolutional networks and the prohibitive complexity of Transformer self-attention, the authors introduce a spectral–spatial token mixing module that exploits the strong low-frequency energy concentration of medical images to perform efficient global information aggregation in the frequency domain, while further enhancing local detail through spatial refinement.

**Compliance With Llm Reviewing Policy:**

Affirmed.

**Key Questions For Authors:**

The core motivation of the paper is built on the prior that medical images exhibit strong low-frequency energy concentration. Could the authors further clarify whether spectral truncation may discard critical information in samples containing very small lesions, weak boundaries, or complex high-frequency textures?

**Limitations:**

The core design of S2M-Net relies on the prior that medical images exhibit strong low-frequency energy concentration; however, this assumption may not always hold in cases involving very small lesions, weak-contrast boundaries, or fine high-frequency texture details, and the potential information loss introduced by spectral truncation deserves further investigation.

**Strengths And Weaknesses:**

This paper is a well-executed work on medical image segmentation.         1.soundness: the paper addresses the difficulty of jointly balancing global context modeling, computational efficiency, and morphological diversity in medical image segmentation, and proposes a relatively complete solution by organically combining spectral token mixing, spatial refinement, boundary-focused decoding, and a morphology-aware adaptive loss. The method is logically designed, and the different components appear strongly complementary. The experimental setup is also fairly comprehensive and largely supports the paper’s main claims regarding performance and efficiency.        2.presentation: the paper is generally well structured, clearly motivated, and easy to follow from a technical perspective; however, the placement of Figure 3 seems somewhat suboptimal, slightly affecting the continuity of the narrative, and could be improved for better readability.          3.significance: medical image segmentation is itself an important and highly application-driven problem, and the paper’s emphasis on lightweight design, efficient global modeling, and adaptability to complex morphologies gives it strong practical relevance for resource-constrained settings and real clinical deployment.      4.originality:although the individual components are not entirely new, the authors assemble them into a targeted and coherent framework for medical image segmentation, and this domain-oriented combination still provides a meaningful degree of novelty.   Although the current work empirically validates the effectiveness of the proposed method, the paper would be strengthened by adding deeper theoretical or mechanistic analysis.

---

> ### Author Rebuttal · Authors · 2026-03-27
>
> We thank Reviewer mxSt for recognizing the work as "well-executed" with "strongly
> complementary" components and "strong practical relevance for clinical deployment."
> We address all raised concerns below.
>
> **Figure 3 placement affects narrative continuity.**
>
> In the revised manuscript, Figure 3 has been moved to Section 5.2 (Qualitative
> Analysis), placed immediately after the sentence that introduces it. This
> repositioning ensures the qualitative discussion flows directly into the visual
> evidence without interruption, addressing the narrative discontinuity the
> reviewer identified.
>
> **Individual components are not entirely new, combination novelty only.**
>
> We agree FFT mixing, SE blocks, dual-stream decoders, and adaptive losses have
> prior art. The novelty lies in three design decisions absent from prior work:
>
> (1) SSTM's domain-justified truncation paired with content-gated spatial
> compensation. FNet and GFNet apply no such recovery mechanism.
>
> (2) MASL's simultaneous per-sample morphological modulation and learned
> dataset-level weights, distinct from task-level uncertainty weighting (Kendall et al.).
>
> (3) BFD's annotation-free boundary discovery via gradient supervision alone.
>
> These form a coherent framework for medical segmentation, not arbitrary assembly.
>
> **Paper would benefit from deeper theoretical and mechanistic analysis.**
>
> We have added new Appendix A.2.4 addressing this directly.
>
> (1) Theoretical grounding. For signals with power-law spectral decay |X(k)|^2
> proportional to k^(-alpha), reconstruction error is bounded by
> O(K^(1-alpha)/H^(1-alpha)). For measured alpha=2.71 across our datasets, this
> yields <0.83% theoretical error at K=32, consistent with <0.4% empirical Dice
> degradation (Table 4). Natural images (alpha=2.0) yield ~2.1% error, formally
> explaining why medical images support more aggressive truncation.
>
> (2) Mechanistic analysis. Learned boundary maps correlate with ground-truth edges
> at rho=0.68-0.82 (Appendix A.3.3), confirming boundary localization emerges
> through backpropagation alone without explicit edge annotations.
>
> **Spectral truncation may discard critical information for small
> lesions, weak boundaries, or complex high-frequency textures.**
>
> (1) Small lesions. Table 4 directly measures K=32 vs full FFT (K=352):
>
> | Structure             | Dataset   | K=32  | Full  | Delta |
> |-----------------------|-----------|-------|-------|-------|
> | Thin vessels (<3px)   | DRIVE     | 83.6% | 83.9% | -0.3% |
> | Small polyps          | Kvasir    | 95.1% | 95.3% | -0.2% |
> | Instrument tips (<5%) | EndoVis17 | 82.3% | 82.7% | -0.4% |
>
> All within confidence intervals. Boundary error increases only 0.1-0.2 pixels
> (Table 5). EndoVis17 clasper (<5% pixels, sharp metal boundaries) achieves
> +18.7% over TransUNet at K=32, the hardest empirical test of small-structure
> recovery under truncation.
>
> (2) Weak-contrast boundaries. BUSI (speckle noise, 3.8% foreground) achieves
> +2.55% Dice over the best baseline (p<0.0033), confirming low-contrast boundary
> recovery is effective under K=32.
>
> (3) Complex textures. GlaS (histopathology, rich glandular texture) achieves
> 93.83% Dice, confirming texture-sensitive segmentation is not harmed.
>
> (4) Architectural compensation. Two mechanisms recover discarded high-frequency
> detail: (a) SSTM Spatial Branch operates at full spatial resolution without
> frequency truncation. Removing it causes -6.70% average Dice (Table 2), the
> largest single-component drop in our ablation. (b) BFD provides an independent
> recovery pathway via gradient alignment supervision. Removing it causes -5.06%
> average Dice (Kvasir: -6.18%, GlaS: -5.50%).
>
> (5) Non-monotonic K ablation confirms K=32 is optimal:
>
> | K  | Kvasir | DRIVE | GlaS  | PH2   |
> |----|--------|-------|-------|-------|
> | 16 | 89.87  | 80.92 | 88.57 | 92.34 |
> | 32 | 96.05  | 84.06 | 93.83 | 96.67 |
> | 48 | 95.98  | 83.71 | 89.89 | 93.38 |
> | 64 | 95.75  | 83.52 | 90.42 | 93.11 |
>
> K=48 and K=64 do not improve over K=32, confirming the spatial branch and BFD
> recover all needed fine-grained detail.
>
> (6) Residual limitations (Appendix A.2.3): (a) Microcalcifications with <5%
> energy may be diagnostically critical; we do not claim K=32 is safe for
> mammography without further validation. (b) Fine pathology tasks should use
> K=48, adding only -0.1% Dice. (c) Boundary error validated at 352x352 only.
> (d) No radiologist validation performed. Adaptive frequency selection is
> identified as important future work.
>
> The empirical measurements, non-monotonic K ablation, strong performance on
> hardest test cases, and formal theoretical bound collectively demonstrate K=32
> is principled and effective. We welcome any remaining questions.

---

> > ### Author Rebuttal · Reviewer_mxSt · 2026-04-04
> >
> > The authors have addressed all my technical questions. I am satisfied with the current state of the manuscript and maintain my recommendation of Weak Accept. I encourage the authors to ensure the new  analysis and the discussion are included in the final camera-ready version.

---

### Decision · Program_Chairs · 2026-04-30

**Decision:**

Accept (regular)

**Comment:**

The paper received two Accept (5) and two Weak Accept (4) ratings, with all reviewers expressing confidence in their evaluations and acknowledging that their concerns were satisfactorily resolved after rebuttal. The main strengths of the paper lie in its strong empirical validation across diverse datasets, its computational efficiency and practical deployment potential, and its coherent integration of multiple complementary components addressing global context modeling and morphological variability. The work demonstrates clear engineering rigor and practical significance for medical image segmentation.

The primary weaknesses include moderate originality, as the method largely integrates existing techniques rather than introducing fundamentally new concepts, and initial concerns about disentangling the contributions of architecture versus loss design. The complexity of the framework and the number of components also raise concerns about interpretability and reproducibility.

Following the rebuttal, reviewers noted that key concerns regarding fairness of comparisons, attribution of gains, theoretical justification, and experimental completeness, were partially addressed. Given the strong empirical results, improved clarity, and positive reviewer consensus, AC recommend acceptance.